# Ferrotoroidic ground state in a heterometallic {Cr$^{III}$Dy$^{III}_6$} complex displaying slow magnetic relaxation

Kuduva R. Vignesh[1], Alessandro Soncini [2], Stuart K. Langley[3], Wolfgang Wernsdorfer[4], Keith S. Murray[5] & Gopalan Rajaraman[6]

Toroidal quantum states are most promising for building quantum computing and information storage devices, as they are insensitive to homogeneous magnetic fields, but interact with charge and spin currents, allowing this moment to be manipulated purely by electrical means. Coupling molecular toroids into larger toroidal moments via ferrotoroidic interactions can be pivotal not only to enhance ground state toroidicity, but also to develop materials displaying ferrotoroidic ordered phases, which sustain linear magneto–electric coupling and multiferroic behavior. However, engineering ferrotoroidic coupling is known to be a challenging task. Here we have isolated a {Cr$^{III}$Dy$^{III}_6$} complex that exhibits the much sought-after ferrotoroidic ground state with an enhanced toroidal moment, solely arising from intramolecular dipolar interactions. Moreover, a theoretical analysis of the observed sub-Kelvin zero-field hysteretic spin dynamics of {Cr$^{III}$Dy$^{III}_6$} reveals the pivotal role played by ferrotoroidic states in slowing down the magnetic relaxation, in spite of large calculated single-ion quantum tunneling rates.

[1] IITB-Monash Research Academy, IIT Bombay, Mumbai 400076, India. [2] School of Chemistry, University of Melbourne, Melbourne, VIC 3010, Australia. [3] School of Science and the Environment, Division of Chemistry, Manchester Metropolitan University, Manchester, M15 6BH, UK. [4] Institute Neel, CNRS, F-38000 Grenoble, France and Institute of Nanotechnology, Karlsruhe Institute of Technology, 76344 Eggenstein-Leopoldshafen, Germany. [5] School of Chemistry, Monash University, Melbourne, VIC 3800, Australia. [6] Department of Chemistry, Indian Institute of Technology Bombay, Mumbai 400076, India. Correspondence and requests for materials should be addressed to A.S. (email: asoncini@unimelb.edu.au) or to K.S.M. (email: keith.murray@monash.edu) or to G.R. (email: rajaraman@chem.iitb.ac.in)

The magnetic behavior of molecular coordination complexes continues to intrigue scientists around the world, revealing many interesting physical properties and offering many potential applications such as new storage and information processing technologies[1–3]. Fundamental research into, e.g., single-molecule magnets (SMMs)[4–6], spin-crossover[7], and magnetic systems with toroidal moments[8–19] are recognized as important areas of molecular magnetism. SMMs exhibit slow relaxation of the magnetization, acting as nano-magnets below their blocking temperature[2, 4–6]. Molecular coordination complexes with a toroidal arrangement of local magnetic moments are rare, but are of great interest as they have several potential applications such as quantum computation, molecular spintronics devices, and the development of magneto–electric coupling for multiferroic materials[20–22]. Toroidal moments at a molecular level were first predicted[12] and observed[9, 17] in 2008, in strongly anisotropic metal complexes with a ring topology, having in-plane magnetic axes tangential to the ring, and weak nearest neighbor exchange coupling of the appropriate sign. In particular, the observation of a toroidal texture[9, 17] in the ground state of the {Dy$^{III}_3$} triangular system[8], generated great interest in this area, with state-of-the-art theoretical[9, 14] and experimental[10, 17, 18] techniques being employed to probe toroidal states of this prototype and subsequent related molecules. Other than triangular {Dy$_3$} complexes, the toroidal moments have been reported as well for rhombus {Dy$_4$}[23–25] and {Dy$_6$} wheel[26] complexes.

If two molecular rings exhibiting toroidal states are connected, e.g., via a 3d ion, then coupling between the two toroidal moments leading to an enhancement of the collective toroidal moment may occur, which is a prerequisite to achieve a molecular ferrotoroidically ordered phase and the development of molecule-based multiferroics[11, 13, 18, 27]. In order to isolate materials with the above mentioned properties, we target coordination complexes containing anisotropic 4f ions. In contrast to the great deal of work on the synthesis of 3d–4f coordination complexes using 3d ions such as Mn$^{III}$, Fe$^{III}$, and Co$^{II}$[28–32], there have been few reports of studies, both structurally and magnetically, on mixed Cr(III)–Ln(III) systems. We have recently shown, however, that the combination of the isotropic Cr$^{III}$ ion and the anisotropic Dy$^{III}$ ion resulted in a family of SMMs with long relaxation times, relative to other lanthanide-based SMMs[33–36]. With this in mind, we have chosen to expand our studies, utilizing chromium(III) nitrate, with various lanthanide(III) ions, with carboxylic acid pro-ligands.

Herein, we describe the synthesis, structural characterization, and magnetic properties of a heterometallic complex of formula [Cr$^{III}$Dy$^{III}_6$(OH)$_8$(*ortho*-tol)$_{12}$(NO$_3$)(MeOH)$_5$]•3MeOH (**1**), where *ortho*-tol = *ortho*-toluate. Complex **1** displays slow magnetic relaxation and SMM behavior at temperatures below 2 K. Furthermore, in {Cr$^{III}$Dy$^{III}_6$} we find, for the first time, a ferrotoroidically coupled ground state fully determined by dipolar coupling between the two con-rotating toroidal triangles. Our observations on **1** have been compared to earlier reported studies on coupled molecular {Dy$^{III}_3$} toroids[10, 11, 18]. The ferrotoroidically coupled ground state thus leads to an enhanced toroidal moment in the ground state for the {Cr$^{III}$Dy$^{III}_6$} complex, which is shown to play a central role in the observed magnetization dynamics featuring zero-field hysteresis.

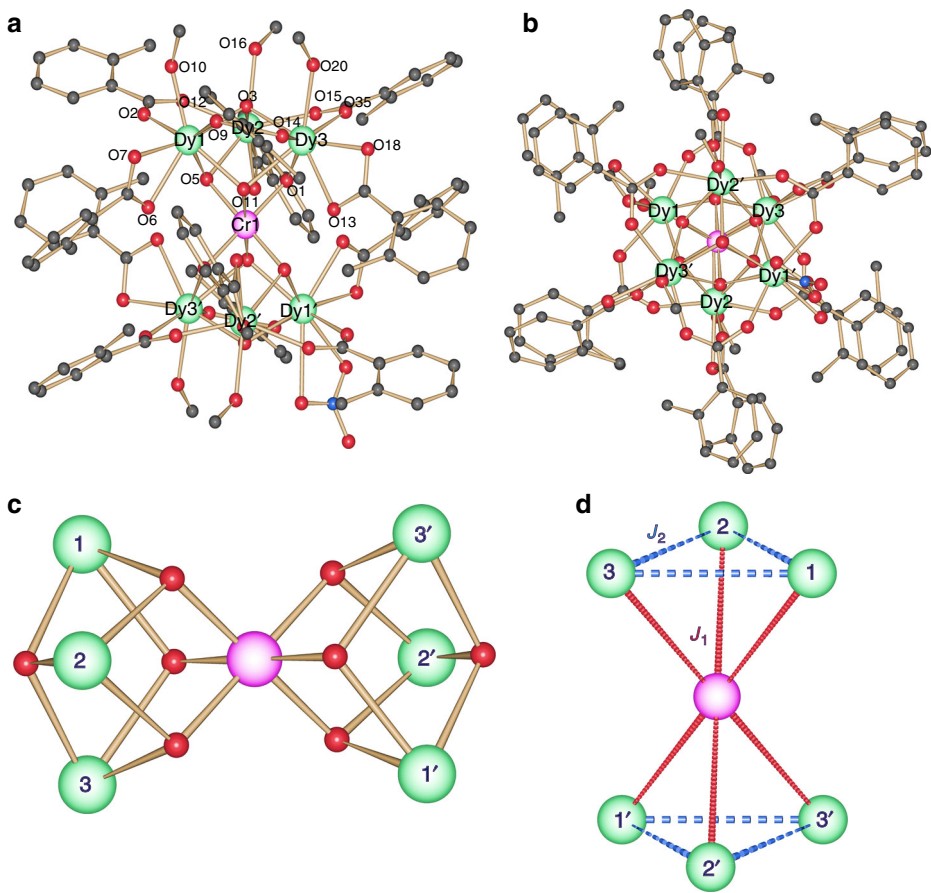

**Fig. 1** Molecular structure and exchange pathways. **a** The molecular structure of complex **1**. The solvent and H atoms are omitted for clarity. Color scheme; Cr$^{III}$, pink; Dy$^{III}$, green; O, red; N, blue; C, light gray; **b** Top view of the molecular structure of **1**; **c** Metal topology found in **1** with **d** magnetic exchange pathways J$_1$, and J$_2$ highlighted

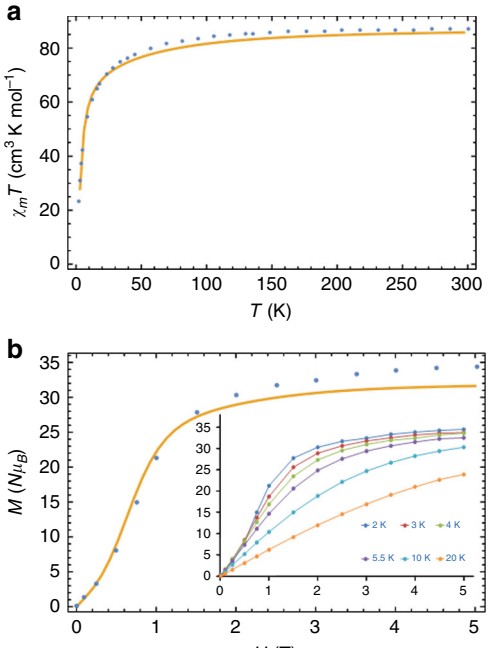

**Fig. 2** Susceptibility and magnetization plots. The measured (blue circles) and simulated (via the ab initio-parameterized model, orange solid line) plot of **a** $\chi_M T$ vs. $T$ at 1 T and **b** $M$ vs. $H$ isotherms at 2 K for complex **1**. (inset) $M$ vs. $H$ isotherms for complex **1** at 2, 3, 4, 5.5, 10, and 20 K (solid lines just join the points here)

## Results

**Synthesis and magnetic properties**. Compound **1** was synthesized by the reaction of $Cr(NO_3)_3\cdot9H_2O$ and $Dy(NO_3)_3\cdot6H_2O$, with *ortho*-toluic acid in acetonitrile at ambient temperature. The solvent was then removed and re-dissolved in $MeOH/^iPrOH$ (Supplementary Methods). Suitable single crystals (pale purple color) for X-ray analysis were isolated after allowing the solvent to slowly evaporate.

Single-crystal X-ray analysis reveals that compound **1** crystallizes in the triclinic space group, $P$-1 (see Supplementary Table 1 for full crystallographic details). The asymmetric unit contains half the complex, (three $Dy^{III}$ ions and one half of the $Cr^{III}$ ion) which lies upon an inversion center. Compound **1** is a heterometallic heptanuclear complex consisting of a single $Cr^{III}$ ion and six $Dy^{III}$ ions (Fig. 1). The low $Cr^{III}$ to $Dy^{III}$ ratio of 1:6 in **1** is likely a consequence of the limited solubility of $Cr(NO_3)_3\cdot9H_2O$ in MeCN. The metallic core is based on two triangular $Dy^{III}$ units that lie above and below a single central $Cr^{III}$ ion, revealing two vertex sharing trigonal pyramids or tetrahedra. The metallic core is stabilized by 8 $\mu_3$-hydroxide, 12 *ortho*-toluate, with MeOH and $[NO_3]^-$ ligands. Six of the $\mu_3$ hydroxide ligands bridge two $Dy^{III}$ ions to the central $Cr^{III}$ ion, while the remaining two bridge the three $Dy^{III}$ ions that make up each triangle. Six of the *ortho*-toluate ligands each bridge a $Dy^{III}$–$Dy^{III}$ triangular edge, while six are found to chelate, each to a single $Dy^{III}$ ion. Terminal MeOH ligands coordinate to all six $Dy^{III}$ ions. It is found, however, that, at two of the $Dy^{III}$ sites, disordered MeOH and nitrate ions (50:50 occupancy) are present. The $Cr^{III}$ ion is six coordinate with an octahedral geometry, while the six $Dy^{III}$ ions are eight coordinate.

We have examined the structural distortions at individual $Dy^{III}$ sites using SHAPE software[37, 38]. The geometry of each $Dy^{III}$ ion is best described by a triangular dodecahedron. The deviation of 2.7 for Dy1 and Dy1′, 1.2 for Dy2 and Dy2′, 1.5 for Dy3 and Dy3′ are observed with respect to the ideal triangular dodecahedron.

Selected bond lengths and $Dy^{III}$–O–$Dy^{III}$ and $Cr^{III}$–O–$Dy^{III}$ bond angles are given in Supplementary Table 2. The $Dy^{III}$–O bond lengths are in the range, 2.391–2.492 Å. In the $\{Dy^{III}_3\}$ triangular unit, the bond distance between Dy1–Dy2, Dy1–Dy3, and Dy2–Dy3 is found to be 3.749, 3.767, and 3.780 Å, respectively and the Dy3–Dy1–Dy2, Dy1–Dy2–Dy3, and Dy2–Dy3–Dy1 bond angles are 55.99°, 60.52°, and 59.49°, respectively. The average Dy-($\mu_3$-OH)-Dy angle is 106.0°, while the average Dy-($\mu_3$-OH)-Dy angle also bridging to the $Cr^{III}$ ion is slightly smaller at 103.4°. The centroid to centroid distance between the two triangular units is found to be 5.38 Å. A shorter distance, compared to other linked $\{Dy^{III}_3\}$ triangles[11] (5.64 Å) is likely to yield stronger dipolar coupling between the two $\{Dy^{III}_3\}$ units. Packing diagrams of **1**, viewed along the $a$, $b$, and $c$ axes and of a neighboring pair are shown in Supplementary Fig. 1. There are H-bonds linking adjacent $\{Cr^{III}Dy^{III}_6\}$ complexes via MeOH····MeOH(solv)···O(carb) groups, combined with edge to face $\pi$–$\pi$ interactions.

Dc (direct current) magnetic susceptibility data were collected for **1** and the variation of $\chi_M T$ with temperature is shown in Fig. 2a. The room temperature $\chi_M T$ product of 87.16 $cm^3$ K $mol^{-1}$ is in agreement with the value expected (86.9 $cm^3$ K $mol^{-1}$) for one $Cr^{III}$ ($S = 3/2$, $g = 2$, $C = 1.875$ $cm^3$ K $mol^{-1}$) and six $Dy^{III}$ ($S = 5/2$, $L = 5$,$^6H_{15/2}$, $g = 4/3$, $C = 14.17$ $cm^3$ K $mol^{-1}$) ions that are non-interacting. As the temperature is reduced the $\chi_M T$ product ($H = 1$ T) decreases gradually between room temperature and 50 K, before a more rapid decrease below this temperature, reaching a value of 14.37 $cm^3$ K $mol^{-1}$ at 1.8 K. The decrease in $\chi_M T$ at higher temperatures is attributed to the depopulation of the excited $m_J$ Stark states of the $Dy^{III}$ ions, while the more rapid decrease at low temperatures is indicative of the presence of dominant antiferromagnetic exchange interactions. The low temperature $\chi_M T$ value of 14.37 $cm^3$ K $mol^{-1}$ at 2 K is higher than that expected for a single paramagnetic Cr(III) ion suggest that there are several close lying excited states including that of $Dy^{III}$ ion, which possess significant magnetic moment.

The isothermal $M$ vs. $H$ plots (Fig. 2b; Supplementary Fig. 2) at low fields reveal a non-linear, S-shaped curve at 2 and 3 K, which suggests the presence of a toroidal moment[8–14] and/or possible blockage of the magnetization vector and therefore slow magnetic relaxation. Above 4 K, the plots display a rapid increase in magnetization up to ~2 T followed by a steady increase and almost saturating at 5 T (Fig. 2b). The simulation of the plots in Fig. 2 are discussed later.

To probe for SMM behavior, the magnetization dynamics were investigated via alternating current (ac) susceptibility measurements as a function of both temperature and frequency. A 3.5 Oe ac field was employed, utilizing both a 0 and a 2000 Oe static dc field. A non-zero out-of-phase magnetic susceptibility component ($\chi''$) is observed, at $H_{dc} = 0$ Oe, however, no maxima are found upon reducing the temperature down to 1.8 K (Supplementary Fig. 3). This is also the case for when $H_{dc} = 2000$ Oe (Supplementary Fig. 3). This does not prove SMM behavior, but suggests the possibility of such, with a small energy barrier to magnetic reorientation and fast relaxation times, even at 1.8 K. As a consequence of the fast magnetic relaxation times, even at temperatures below 2 K, it is suggested that the low-field magnetization behavior points to the presence of a toroidal magnetic moment.

Single crystals of **1** were studied using the micro-SQUID technique at various temperatures and sweep rates for two different orientations of the molecule[39]. The curve displays a stepped shape of the magnetization (Fig. 3; Supplementary Fig. 4), similar to that observed by the archetypal triangular toroidal system[8]. The position of the step ($H_s$) depends on the orientation of the magnetic field with respect to the plane of the triangles, as

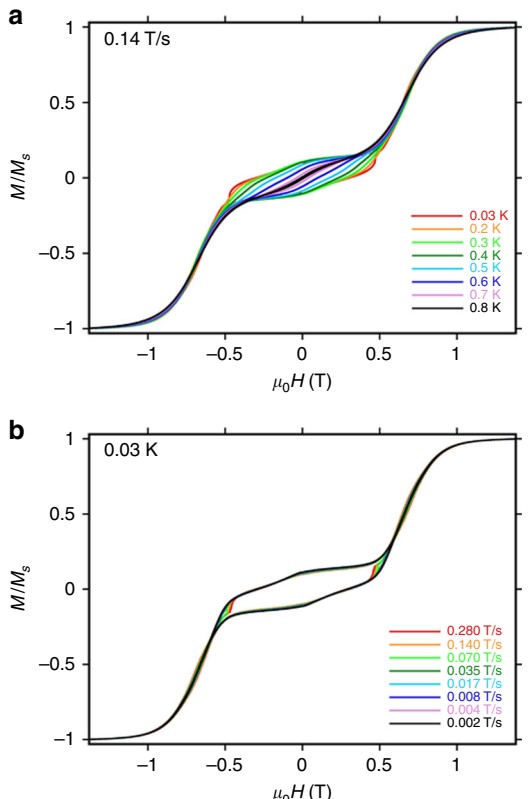

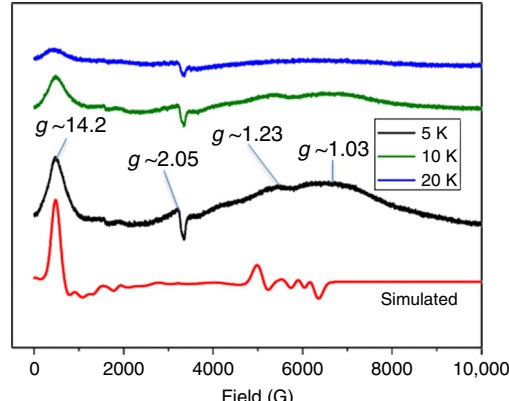

**Fig. 4** EPR spectroscopy. Powder EPR spectra of **1** at X-band frequency at 5, 10, and 20 K and the simulated curve

**Fig. 3** Single-crystal studies. Single-crystal magnetization (*M*) vs. applied field measurements (μ-SQUID) for complex **1** at **a** 0.03–0.8 K with the scan rate of 0.14 T s$^{-1}$; **b** with different field sweep rates at 0.03 K

expected on the basis of the in-plane anisotropy predicted by our model (vide infra). Thus, when the field is parallel to two inversion-related tangential Dy magnetic axes, and perpendicular to none, this leads to an in-plane easy axis, while a magnetic field perpendicular to that direction, thus perpendicular to two Dy easy axes and parallel to none, leads to an in-plane hard axis. In particular, when the field is applied along any Dy–Dy bond vector as in Fig. 3a, i.e., along the y-axis in Fig. 6, hysteresis is observed below 0.8 K, with the coercive field widening on cooling ($H_c = 0.6$ T at 0.03 K, with a sweep rate of 0.28 T s$^{-1}$). This behavior is characteristic of an SMM, with slow zero-field relaxation. We then observe a large step in the magnetization at about $H_s = 0.7$ T. Application of the magnetic field perpendicular to the Dy–Dy bond vector results in a reduction of coercivity ($H_c = 0.5$ T at 0.03 K, with a sweep rate of 0.28 T s$^{-1}$, Supplementary Fig. 4). Upon comparison of the hysteresis profile of complex **1** with the archetypal {Dy$_3$} toroidal complex[8], we find that profile for **1** appears to be superior comparing the coercitivity. This suggests that coupling between the two {Dy$^{III}_3$} triangles in **1** enhances the zero-field slow relaxation properties of the system. A theoretical model explaining this behavior is developed later in the paper.

We have also recorded EPR spectra at the X-band frequency at 5, 10, and 20 K (Fig. 4). The EPR spectrum recorded at 5 K reveals distinct features at very large *g*-values (*g* ∼ 14.2). When we increased the temperature we found that the intensity of this signal decreases. There are also weak features at *g* ∼ 2.05, *g* ∼ 1.2, and *g* ∼ 1.03 and the intensities of these features also decrease upon increasing the temperature. To gain an understanding on the nature of these EPR spectra, we have simulated the EPR spectrum by means of the XSOPHE simulation suite[40, 41], using a {Cr$^{III}$Dy$^{III}_3$} model employing a pseudo $S = 1/2$ state for each

Dy$^{III}$ ion and a $S = 3/2$ state for the Cr$^{III}$ ion. Ab initio computed *g*-anisotropies, directions, and *J* values are given as inputs (see below for details) along with the Dy···Dy and Dy···Cr distances from the X-ray structure. A small perturbation to the Euler angles without altering any other parameters yield reasonable fit to the experimental spectrum recorded at 5 K (Fig. 4; The further details of simulation are given in Supplementary Note 1), offering confidence on the estimated parameters. However, the lines appearing at *g* ∼ 1.23 and *g* ∼ 1.03 are much broader than they appear in the simulation and this may be attributed to a fact that only {Cr$^{III}$Dy$^{III}_3$} has been employed in the simulation and not the full {Cr$^{III}$Dy$^{III}_6$} Hamiltonian. Multi-frequency EPR including HF-EPR spectra are required, in future, to independently obtain the spin Hamiltonian parameters[42].

**Theoretical analysis and characterization of a ferrotoroidic ground state**. To explain the experimental observations, we performed ab initio calculations of electronic structure and magnetic properties, using MOLCAS 7.8[43], on individual Dy$^{III}$ and Cr$^{III}$ centers. The computed orientation of the anisotropy axes is shown in Fig. 5. In particular, we employed the ab initio $M_J$ decomposition of the single-ion thermally isolated ground Kramers doublet (KD) wavefunctions along the ab initio *g*-tensor principal axis, to set up a model Hamiltonian for intramolecular magnetic coupling including dipolar coupling between all pairs of ions, which is parameters-free, intra-ring Dy$^{III}$–Dy$^{III}$ superexchange interactions parameterized by a single coupling constant $J_2$, and Dy$^{III}$–Cr$^{III}$ superexchange interactions parameterized by a single coupling constant $J_1$. The coupling parameters $J_1$ and $J_2$ were evaluated via DFT calculations. The well-known dipolar Hamiltonian reads:

$$H_{dip} = \frac{\mu_0}{4\pi} \sum_{p,q} \left( \frac{\mathbf{M}_p \cdot \mathbf{M}_q}{|\mathbf{R}_{pq}|^3} - 3\frac{(\mathbf{M}_p \cdot \mathbf{R}_{pq})(\mathbf{M}_q \cdot \mathbf{R}_{pq})}{|\mathbf{R}_{pq}|^5} \right), \quad (1)$$

where $\mathbf{M}_p$ is the magnetic moment of the *p*th ion, and $\mathbf{R}_{pq}$ the distance between ions *p* and *q*. The superexchange contribution is modeled by an isotropic Heisenberg Hamiltonian[44]:

$$H_{exch} = -J_2 \sum_{p,q} (S_p \cdot S_q + S_{p'} \cdot S_{q'}) - J_1 \sum_q (S_q + S_{q'}) \cdot S_{Cr}, \quad (2)$$

where $\mathbf{S}_q$ ($\mathbf{S}_{Cr}$) are the true spin moments of the Dy (Cr) ions, with the primed and unprimed subscripts labeling Dy ions belonging to different triangles. We note here that when the simple isotropic exchange Hamiltonian is projected on the thermally isolated ground KDs, it becomes a strongly anisotropic

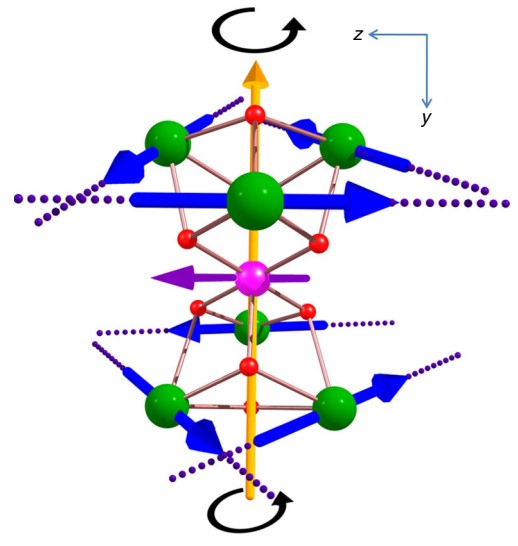

Fig. 5 Orientation of the magnetic anisotropy axes. The directions of the local anisotropy axes in the ground Kramers doublet on each Dy site (dotted lines) in **1**. Blue arrows are the local magnetic moment in the ground exchange doublet. Black arrows show the con rotation of the toroidal magnetic moment and the yellow arrow is the $S_6$ symmetry axis

non-collinear Ising Hamiltonian, a widely employed protocol for {3d–4f} systems.

To estimate the low-energy wavefunctions and magnetic anisotropy for each of the seven ions in **1**, we have undertaken CASSCF+RASSI-SO calculations on the individual Dy$^{III}$ centers[43]. The calculations yielded the following g-tensor principal values: (Dy1; $g_x = 0.0523$, $g_y = 0.0927$, and $g_z = 19.5707$); (Dy2; $g_x = 0.0737$, $g_y = 0.0979$, and $g_z = 19.4723$); (Dy3; $g_x = 0.0233$, $g_y = 0.0361$, and $g_z = 19.6059$) and (Cr; $g_x = g_y = g_z = 2.002$) (Supplementary Tables 3, 6). The symmetry-related Dy1′, Dy2′, and Dy3′ ions possess essentially the same g-tensor. Although the two triangles are equivalent, Dy1 and Dy1′ slightly differs due to coordination of methanol in Dy1 and nitrate in Dy1′. These data, along with the $J$ value are found to yield a good fit to the experimental susceptibility data (Fig. 2, vide supra). A qualitative mechanism developed based on single-ion Dy(III) is discussed in detail in the ESI (Supplementary Fig. 6; Supplementary Note 2).

The core structure of the molecule has a pseudo $S_6$ axis passing through the Cr$^{III}$ ion and the center of both of the {Dy$^{III}_3$} triangular units (Fig. 5). The local principal anisotropy axes are found from the calculations to lie in the {Dy$^{III}_3$} plane with an out-of-plane angle of 0.29, 4.5, and 4.7° for Dy1, Dy2, and Dy3, respectively. The Dy$^{III}$ magnetic axes are also found to be almost perfectly aligned with the tangents to an ideal circumference enclosing the triangles (the angle of the anisotropy axis with these tangential directions are in the range of 1.1–7.9°). The computed energies of the eight low-lying KDs reflect that there are three types of Dy$^{III}$ ions in the complex (Supplementary Tables 4, 5), although the ground states of all Dy$^{III}$ ions consist of almost pure atomic $|\pm 15/2>$ KDs. The energy gap between the ground and the first excited state KDs are found to be 142.8, 121.9, and 152.7 cm$^{-1}$ for Dy1, Dy2, and Dy3, respectively. For the Cr$^{III}$ ion calculations yield isotropic g-tensors (Supplementary Table 3) and an axial zero-field splitting of $\sim 1.0 \times 10^{-4}$ cm$^{-1}$.

Thus the ab initio calculations suggest that magnetic coupling can be well described by projecting Hamiltonians Eqs. (1) and (2) on the basis of the ground KDs (Dy) and quartet (Cr) only, leading to a 256-dimensional product space with basis $|\mathbf{m}, M_{Cr}\rangle$,

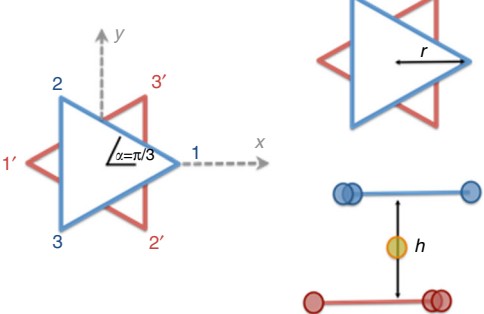

Fig. 6 Schematic diagram to describe exchange coupling. Schematic representation of the idealized geometry used to describe magnetic coupling in {Dy$_6$Cr} via Hamiltonians Eqs. (1) and (2)

where $\mathbf{m} = [m_1, m_2, m_3, m_{1'}, m_{2'}, m_{3'}]$ with $m_q = \pm 1$. We further assume that such KDs are pure $|\pm 15/2>$ atomic states quantized along the local anisotropy axis, which is assumed to be exactly in the triangle's plane and along the tangential direction, so that $\hat{S}_{t,q}|m_q\rangle = m_q(5/2)|m_q\rangle$, and $\hat{J}_{t,q}|m_q\rangle = m_q(15/2)|m_q\rangle$ ($\hat{S}_{t,q}$ and $\hat{J}_{t,q}$ are the spin and total angular momenta operators along the tangential direction for the qth Dy$^{III}$ ion), and a seventh quantum number $M_{Cr} = \pm 3/2, \pm 1/2$ labeling the spin state for the Cr ion. Finally, given the quasi $S_6$ symmetry, in our model we assume two equilateral triangles with radius $r = 2.17$ Å, their planes being at distance $h = 5.38$ Å (Fig. 6, $r$ and $h$ are average experimental values). Thus a ferrotoroidic (FT) state (con-rotating toroidal moments $\pm \tau_1, \pm \tau_2$ on the two triangles) correspond to $|\pm 1, \pm 1, \pm 1, \pm 1, \pm 1, \pm 1, M_{Cr}\rangle \equiv |\pm \tau_1, \pm \tau_2, M_{Cr}\rangle$, while antiferrotoroidic (AFT) states (counter-rotating toroidal moments) correspond to $|\pm 1, \pm 1, \pm 1, \mp 1, \mp 1, \mp 1, M_{Cr}\rangle \equiv |\pm \tau_1, \mp \tau_2, M_{Cr}\rangle$.

Due to the large value of the ground KDs, angular momenta projections in $|\mathbf{m}, M_{Cr}\rangle$, Hamiltonians Eqs. (1) and (2) are both diagonal on such basis, and $|\mathbf{m}, M_{Cr}\rangle$ represent the low-energy exchange-coupled states of **1**. The corresponding energies can be written as a sum of a superexchange contribution, an intra-ring dipolar contribution, an inter-ring dipolar contribution, and Dy$^{III}$–Cr$^{III}$ dipolar contribution.

For the exchange energies we get (sum over q is understood modulus 3, so that $m_{3+1} = m_1$):

$$E_{exch,\mathbf{m}} = \frac{25}{8} J_2 \sum_{q=1}^{3} \left( m_q m_{q+1} + m_{q'} m_{q'+1} \right) - J_1 M_{Cr} \mathcal{M}_{\mathbf{m}}, \quad (3)$$

where defining the angular coordinates of the six Dy$^{III}$ ions as in Fig. 6 ($\alpha_1 = 0$, $\alpha_2 = 2\pi/3$, $\alpha_3 = 4\pi/3$, $\alpha_{1'} = \pi$, $\alpha_{2'} = -\pi/3$, $\alpha_{3'} = \pi/3$), the total spin projection ($\mathcal{M}_{\mathbf{m}}$) of the six Dy$^{III}$ ions in $|\mathbf{m}, M_{Cr}\rangle$ is given by Eq. (4) (sum is over all Dy$^{III}$ centers):

$$\mathcal{M}_m = \pm \frac{5}{2} \sqrt{\left( \sum_q m_q \sin \alpha_q \right)^2 + \left( \sum_q m_q \cos \alpha_q \right)^2} \quad (4)$$

along the direction given by the vector $\mathbf{u}_m = \left( -\sum_q m_q \sin \alpha_q, \sum_q m_q \cos \alpha_q, 0 \right)$ (i.e., lying in the triangle's planes).

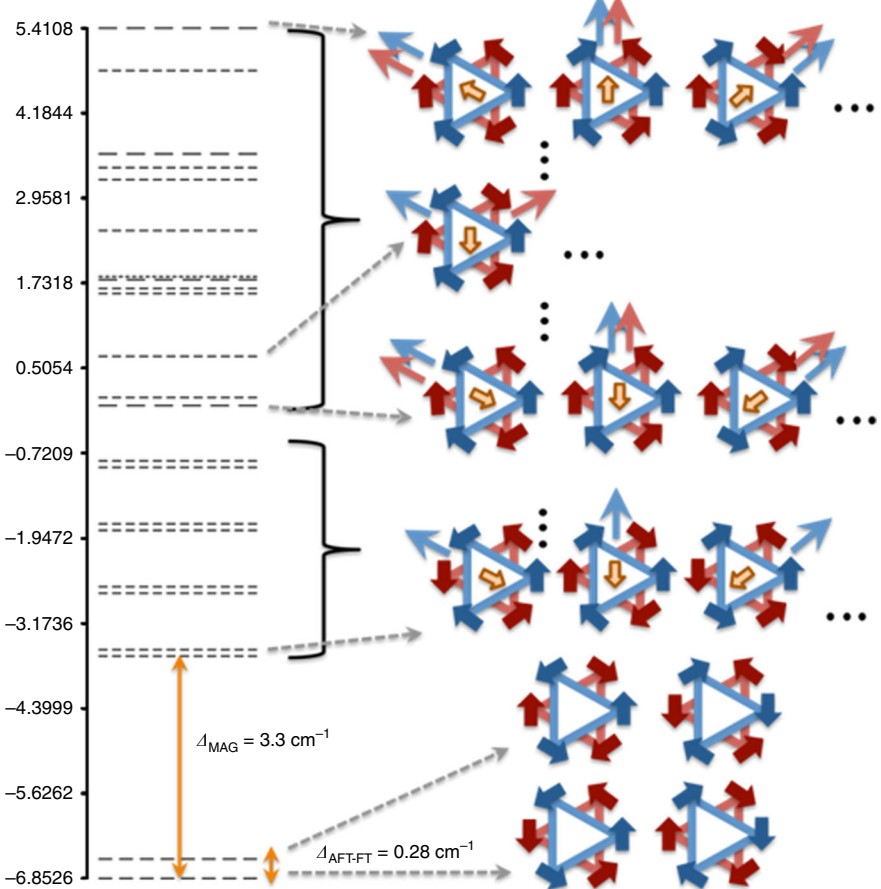

**Fig. 7** Energy spectrum describes magnetic relaxation and toroidal states. The Energy spectrum of {Dy$_6$Cr} calculated using the energy expressions Eqs. (3)–(7). Energies are in cm$^{-1}$. The number of dashes for each degenerate energy level indicates the number of degenerate states associated to that level. Besides a few energy levels, it provides a graphical representation of one of the corresponding non-collinear Ising quantum states $|\mathbf{m}, M_{Cr}\rangle$, where the red (blue) thick arrows at the Dy$^{III}$ sites indicate a +15/2 or −15/2 quantum number along the tangent direction in the lower (upper) triangular ring, a yellow thick arrow at the center indicates the spin projection of the Cr$^{III}$ ion relative to that of the Dy$^{III}$ ions, and a resulting magnetic moment in the upper (lower) ring is indicated with thin blue (red) arrow. Gap between the ground FT state $|\pm\tau_1, \pm\tau_2, M_{Cr}\rangle$, and the first (AFT) $|\pm\tau_1, \mp\tau_2, M_{Cr}\rangle$ excited state are indicated

Intra-ring (Eq. (5)) and inter-ring (Eq. (6)) dipolar coupling energies for the state $|\mathbf{m}, M_{Cr}\rangle$ are ($\mu_B$ is the Bohr magneton):

$$E_{\text{dip},\mathbf{m}}^{\text{intra}} = -\frac{\mu_0\mu_B^2}{4\pi}\frac{125\eta^2}{3\sqrt{3}r^3}\sum_{q=1}^{3}\left(m_q m_{q+1} + m_{q'} m_{q'+1}\right), \quad (5)$$

$$E_{\text{dip},\mathbf{m}}^{\text{inter}} = \frac{\mu_0\mu_B^2}{4\pi}\frac{25\eta^2}{(r^2+h^2)^{\frac{3}{2}}}\left[2-\frac{9r^2}{r^2+h^2}\right]\sum_{q=1}^{3}m_q\left(m_{q+1}+m_{q'-1}\right)$$
$$-\frac{\mu_0\mu_B^2}{4\pi}\frac{100\eta^2}{(4r^2+h^2)^{3/2}}\sum_{q=1}^{3}m_q m_{q'}. \quad (6)$$

Finally, Dy$^{III}$–Cr$^{III}$ dipolar coupling energy for $|\mathbf{m}, M_{Cr}\rangle$ reads:

$$E_{\text{dip},\mathbf{m}}^{\text{Dy–Cr}} = -\frac{\mu_0\mu_B^2}{4\pi}\frac{40\eta}{(r^2+h^2/4)^{3/2}}M_{Cr}\mathcal{M}_{\mathbf{m}}. \quad (7)$$

The energy spectrum resulting from magnetic coupling can now be evaluated summing up Eqs. (3)–(7), and is reported in Fig. 7.

The dipolar coupling energies, Eqs. (5)–(7), contain no free parameter, as they only depend on the specific set of quantum

numbers $|m, M_{Cr}\rangle$, on the experimental geometrical parameters $r$ and $h$ (Fig. 6), and on the average ground state ab initio magnetic moment given by $10\eta\mu_B$, with $\eta = 0.975$. Interestingly, from Eq. (5) it follows that intra-ring dipolar interactions always favor a toroidal texture on each triangle, penalizing the formation of a magnetic moment by an energy gap of ~4 cm$^{-1}$.

While inter-ring dipolar coupling is smaller than intra-ring coupling, due to larger separation between Dy$^{III}$ ions, from Eq. (6) we learn that in fact this interaction splits FT and AFT states. In particular, the first term in Eq. (6) stabilizes a FT (AFT) ground state if $h<r\sqrt{7/2}$ ($h>r\sqrt{7/2}$), while the second term (describing interactions between inversion-related centers) always favors FT coupling, and is stronger than the first term. In {Cr$^{III}$Dy$^{III}_6$}, $h>r\sqrt{7/2}$, hence the two terms are in competition, the first (second) favoring AFT (FT) coupling. Since the second term is larger, we find here that dipolar interactions stabilize a FT ground state in {Cr$^{III}$Dy$^{III}_6$} (Fig. 5).

Moreover, from Eq. (6), we note that a structural design aimed at reducing the distance $h$ between the two triangles will enhance FT coupling. We estimate the dipolar-induced FT/AFT splitting to be ~0.28 cm$^{-1}$. Note that, in our symmetric model, Dy–Cr dipolar interactions within FT and AFT states (Eq. (7)) are exactly zero (since $\mathcal{M}_{\mathbf{m}} = 0$ in FT and AFT states), leading to eight-fold degenerate FT and AFT manifolds. Thus in FT and

AFT states, the spin of Cr(III) is freely fluctuating, and FT/AFT toroidal excitations, involving the collective flipping of three $Dy^{III}$ spins, are fully determined by dipolar coupling. The energy gap of $3.3\,cm^{-1}$ reported in Fig. 7 corresponds instead to the lowest magnetic excitation, obtained upon flipping a single $Dy^{III}$ spin. In the excited states, the Cr(III) spin is blocked in the direction of the in-plane $Dy_6$ magnetic moment.

Considering now superexchange interactions in Eq. (3), we note that the dipolar-induced FT ground state will survive provided that the intra-ring Dy–Dy coupling is antiferromagnetic (i.e., $J_2 < 0$), or ferromagnetic but smaller than dipolar coupling. To estimate the two exchange coupling constants $J_1$ and $J_2$ appearing in Eq. (3), we have employed DFT calculations, replacing the $Dy^{III}$ with $Gd^{III}$ ions in the X-ray structure. The computed coupling constants for the $Cr^{III}$–$Gd^{III}$ pairs were then rescaled by the ratio between the spin of $Dy^{III}$ ($S = 5/2$) and that of $Gd^{III}$ ($S = 7/2$), while $Gd^{III}$–$Gd^{III}$ were rescaled by the ratio between the square of $S = 5/2$ and the square of $S = 7/2$[33, 34] We obtained $J_1 = -0.08\,cm^{-1}$ ($Cr^{III}$–$Dy^{III}$ coupling) and $J_2 = -0.043\,cm^{-1}$ (intra-ring $Dy^{III}$–$Dy^{III}$ coupling), indicating antiferromagnetic coupling. The estimated antiferromagnetic interaction within the $\{Dy^{III}_3\}$ triangles reinforces the effect of the intra-ring dipolar coupling, leading to a toroidal moment on each isolated triangle.

While a degenerate FT quantum ground state is compatible with the inversion symmetry of the molecule, such symmetry is not compatible with a ferrotoroidically ordered phase. Thus upon FT ordering, a concomitant structural phase transition should occur to rid the crystal of the inversion center, which, in turn, would allow the appearance of linear magneto–electric coupling.

We now use the developed theoretical model to simulate the experimental molar susceptibility, and magnetization. We report the results in Figs. 2 and 8. While the magnetization at 2 K and for fields up to 5 T is expected to be dominated by the low-energy states described by our model (Eqs. (1) and (2)), the molar susceptibility will be dominated by these states only at low temperatures, while at temperatures much higher than the coupling strength it will be dominated by the single-ion response, including the population of excited KDs. To reproduce the correct temperature dependence of $\chi_M T$ within our non-empirical model, we, therefore, evaluated the molar susceptibility as $\chi T = (\chi T)_{LT} + (\chi T)_{HT} - (\chi_0 T)$, where $(\chi T)_{LT}$ is computed from our model Eqs. (1) and (2), $(\chi T)_{HT}$ is computed as a sum of the ab initio susceptibilities of the six $Dy^{III}$ ions and the central $Cr^{III}$ ion, while $(\chi_0 T) = 0.125 \times (6M(Dy)^2 + 4 \times 3/2\ (3/2 + 1))$ is the uncoupled contribution to the Curie molar susceptibility ($cm^3\,K\,mol^{-1}$) arising from the six $Dy^{III}$ ground KD's using the ab initio magnetic moment $M(Dy) = 9.75\mu_B$ and the ground quartet on $Cr^{III}$. The results of the calculations of $\chi T$ compared with the experimental data are reported in Fig. 2a (orange line), which shows an excellent agreement between theory and experiment.

Note also that despite the absence of fitting parameters in our model, the theoretically calculated magnetization curve reported in Fig. 2b and Supplementary Fig. 7 (orange curve) reproduces very well the experimentally measured $M$ vs. $H$ curve at 2 K and higher temperatures (blue data points). In particular, the predicted FT and AFT low-energy states display a strongly reduced magnetic response, as is evident from the S-shape behavior of the $M$ vs. $H$ curve. A uniform magnetic field does not interact with a toroidal moment, so that only the $Cr^{III}$ ion responds to the field for low $H$. As the Zeeman energy increases, at $H \sim 0.4$–0.5 T, a level crossing occurs (vide infra), corresponding to a magnetic excited state becoming the ground state, explaining the S-shape of the $M$ vs. $H$ curve. Finally we present a detailed comparison in the Supplementary Note 3 and 4 where various possible models are discussed and our findings are

compared with previous studies of coupled molecular $\{Dy^{III}_3\}$ toroids (Supplementary Fig. 9)[10, 11, 18].

**Theoretical analysis of the zero-field hysteretic spin dynamics.** As a first attempt to interpret the single-crystal magnetization reported in Fig. 8a (blue curve), measured at $T = 0.03$ K and a field sweep rate of $0.1\,T\,s^{-1}$, we calculated the equilibrium magnetization at the same temperature and magnetic field orientation (along the easy axis, $y$-direction in Fig. 6), also reported in Fig. 8a (orange curve). Experimental and theoretical curves share a few common features, both displaying a rise of the magnetization for intermediate fields, separating two magnetization plateaus corresponding to low- and high-field regions. Theoretical and experimental plateaus are seen to coincide, with the theoretical low-field plateau corresponding to saturation of the free fluctuating Cr spin ($M_{SAT} \sim 3\mu_B$) within the FT ground state. According to the Zeeman spectrum in Fig. 8b, c, the steep magnetization step observed in the theoretical magnetization at the level crossing field $B_{LC} \sim 0.4$ T can be identified with the switching of the ground state from the low-field weakly magnetic FT Zeeman state $|\pm\tau, \pm\tau\rangle \equiv |\pm 1 \pm 1 \pm 1 \pm 1 \pm 1 \pm 1, -3/2\rangle$ (Supplementary Fig. 8a), to the high-field onion magnetic state $|+m, +m\rangle \equiv |+1 + 1 - 1 - 1 + 1 + 1, +3/2\rangle$, in which half of the Dy spins circulate clockwise, the other half anticlockwise (Supplementary Fig. 8d), adding up to a magnetic moment $M \sim 40\mu_B$ polarized along the field.

As expected, the calculated equilibrium magnetization cannot reproduce inherently dynamical features observed in the experiments, as in the calculations transitions between energy levels are instantaneous on the timescale of the experiment. There are, in particular, two important differences between theory and experiment displayed in Fig. 8a: in the experiment, a hysteresis loop opens up around the zero-field region between $-B_{LC}$ and $B_{LC}$, and the experimental magnetization step separating low- and high-field plateaus is not abrupt, but occurs within $\sim 0.5$ T field range starting at $B_{LC}$.

Thus, to further analyze the measured magnetization dynamics, we explicitly consider finite-rate transitions between Zeeman states. We are not interested in the microscopic details of such transitions, other than these are driven by terms in the Hamiltonian that have so far been neglected (either because small, or because describing coupling with the surroundings), and other than broadly separating such processes in tunneling transitions characterized by coupling constants $\gamma_i$, active between degenerate energy levels, and phonon-induced transitions characterized by coupling constants $\Gamma_i$, from higher to lower-energy states, with a probability that grows as the cube of the energy gap. The coupling constants $\gamma_i$ and $\Gamma_i$ are related to the matrix elements of the relevant Hamiltonian between initial and final states.

Crucially, we propose that a hierarchy should exist in the magnitude of $\gamma_i$ and $\Gamma_i$, based on the number of $Dy^{3+}$ spins that need to be flipped to change initial to final Zeeman states. We only consider in our model the 56 (out of 256) most relevant low-energy states involved in the relaxation dynamics, which are represented in Supplementary Fig. 8. Particularly relevant are the low-field ground FT states $|\pm\tau, \pm\tau\rangle$, the high-field ground onion state $|+m, +m\rangle$, and the excited intermediate magnetic states $|\pm\tau, +m\rangle$ and $|+m, \pm\tau\rangle$ (Supplementary Fig. 8c). Let us label the coupling constants $\gamma_i$ ($\Gamma_i$) as: $\gamma_{Cr}$ ($\Gamma_{Cr}$) for transitions that only flip the $Cr^{3+}$ spin; $\gamma_1$ ($\Gamma_1$) for transitions flipping only one $Dy^{3+}$ spin; $\gamma_2$ ($\Gamma_2$) for transitions involving two simultaneous spin-flipping events; $\gamma_3$ ($\Gamma_3$) for transitions involving three or more spin-flipping events. Thus the hierarchy we invoke here reads: $\gamma_{Cr} \gg \gamma_1 > \gamma_2 > \gamma_3 \sim 0$, and $\Gamma_{Cr} \gg \Gamma_1 > \Gamma_2 > \Gamma_3 \sim 0$.

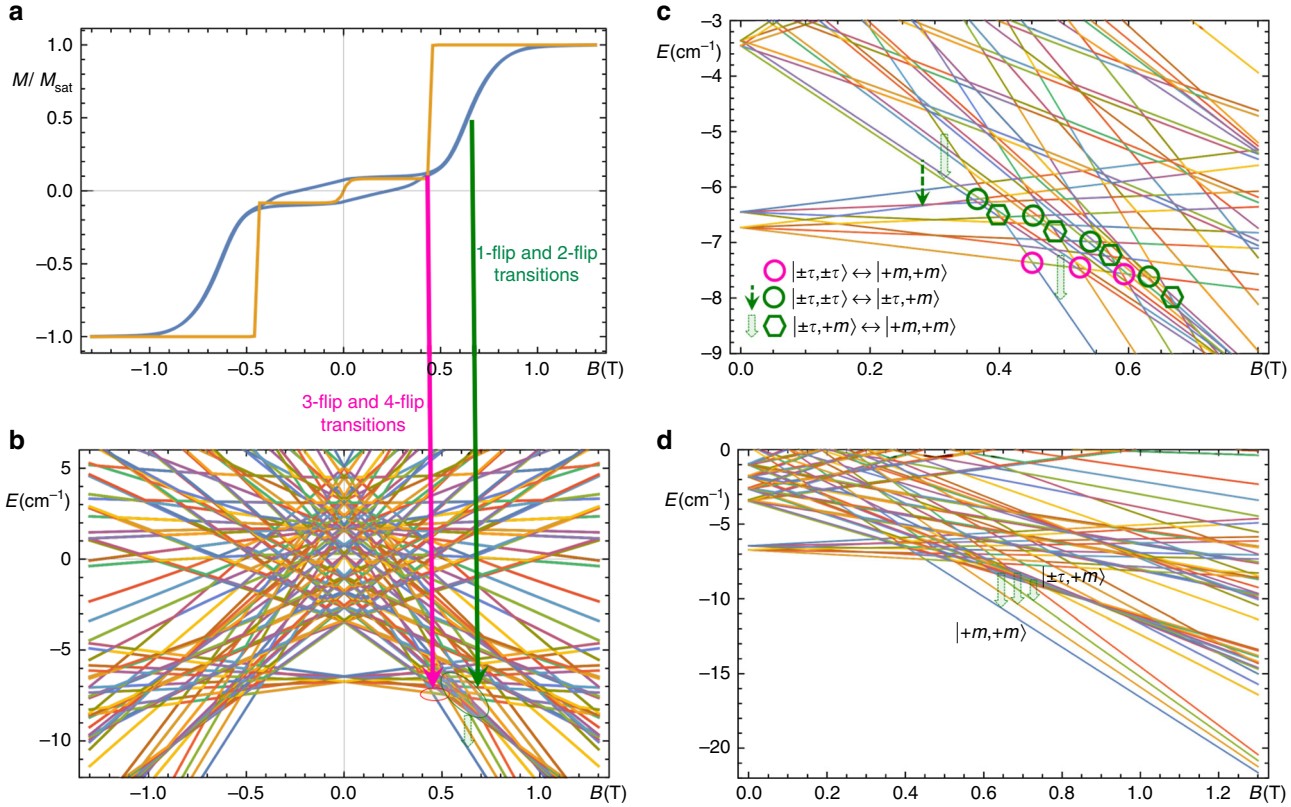

**Fig. 8** Interpretation of single-crystal magnetization experiments. **a** Single-crystal magnetization experiment (blue curve) measured at $T = 0.03$ K and a sweep rate of 0.1 T s$^{-1}$ for a magnetic field oriented along the in-plane easy axis ($y$-axis in Fig. 6), superimposed on the theoretical equilibrium magnetization (orange curve) computed for the same temperature and field direction; **b** Energy levels as function of magnetic field (Zeeman spectrum) computed using the model presented in the text. The magenta (green) arrows connecting **a** and **b** associate the steep (smooth) raise of the theoretical (experimental) magnetization with slow/quasi-forbidden (fast/allowed) transition mechanisms occurring at level crossing between the two ground states (involving excited states) encircled in magenta (green). **c** Magnification of the low-field region of the Zeeman spectrum, highlighting degeneracy points (level crossings) of states between which faster 1-flip and 2-flip tunneling transitions are allowed (green circles and hexagons), and also highlighting 3-flip or higher-flip processes that are essentially forbidden (magenta circles). Some of the allowed phonon emission processes are also indicated with dashed green arrows; **d** High-field region of the Zeeman spectrum, where the four onion states antiferromagnetically coupled to Cr spin states become isolated from excited states, leading the system's magnetization to saturate

Under such assumptions, we argue that both the zero-field hysteresis, and the smooth magnetization step observed in the experiment, can be explained by the fact that the direct $|\pm\tau, \pm\tau\rangle \leftrightarrow |+m, +m\rangle$ transition between the Zeeman ground states at level crossing $B_{LC} \sim 0.4$ T is essentially forbidden, as it involves at least the simultaneous flipping of three Dy$^{3+}$ spins ($\gamma_3$, $\Gamma_3 \sim 0$). Hence, on the timescale of field sweeping, exchange of population between $|\pm\tau, \pm\tau\rangle$ and $|+m, +m\rangle$ can only be indirect, occurring via multi-step processes involving the excited intermediate states $|+m, \pm\tau\rangle$ and $|\pm\tau, +m\rangle$, also via the excited AFT states $|\pm\tau, \mp\tau\rangle$ (Supplementary Fig. 8b). This microscopic scenario is visualized in Fig. 8c, where a multitude of level crossings of states between which faster 1-flip and 2-flip transitions can occur are highlighted with green circles (tunneling) or green dashed arrows (phonon-mediated) ($|\pm\tau, \pm\tau\rangle \leftrightarrow |\pm\tau, +m\rangle$ or $|\pm\tau, \pm\tau\rangle \leftrightarrow |+m, \pm\tau\rangle$), or highlighted with green hexagons (tunneling) and broad green arrows with a dashed contour (phonon-mediated) ($|\pm\tau, +m\rangle \leftrightarrow |+m, +m\rangle$ or $|+m, \pm\tau\rangle \leftrightarrow |+m, +m\rangle$). As illustrated in Fig. 8b, c, the existence of a broad range of magnetic fields around $B_{LC} \sim 0.4$ T for which fast transitions $|\pm\tau, \pm\tau\rangle \leftrightarrow |\pm\tau, +m\rangle$ can occur, followed, in higher fields, by fast phonon relaxation from higher-energy intermediate to lower-energy onion states (Fig. 8d), provides a rationalization for the gradual rise of the experimental dynamical magnetization.

Moreover, to quantitatively investigate the origin of the observed zero-field hysteresis loop, we implement these ideas in a dynamical model based on generalized Pauli master equations describing the dissipative dynamics of the non-equilibrium thermal populations of the CrDy$_6$ states, coupled to an equilibrium reservoir of acoustic phonons at $T = 0.03$ K, and a source of random stray fields inducing incoherent tunneling between resonant energy levels. The states are obtained from our model Hamiltonians Eqs. (1) and (2), also including a time-dependent Zeeman term describing the interaction with the sweeping magnetic field-oriented along the easy axis, at sweeping rate 0.1 T s$^{-1}$, varying between $B = \pm 1.4$ T (a triangle wave signal is used). As outlined in the methods section, the relevant dissipative equations of motion for diagonal elements $\sigma_{ii}$ (populations) of the reduced density matrix $\sigma$, in the adiabatic approximation, are:

$$\dot{\sigma}_{ii}(t) = \sum_k \left\{ W_{k \to i}^{ph}(t) + \Omega_{k \leftrightarrow i}^{tun}(t) \right\} \sigma_{kk}(t)$$
$$- \sigma_{ii}(t) \sum_k \left\{ W_{i \to k}^{ph}(t) + \Omega_{k \leftrightarrow i}^{tun}(t) \right\},$$

(8)

where $\Omega_{k \leftrightarrow i}^{tun}(t)$ and $W_{i \to j}^{ph}(t)$ are the time-dependent tunneling and phonon-induced transition rates given by Eqs. (12) and (10), respectively, proportional to the coupling constants $\gamma_i$ and $\Gamma_i$

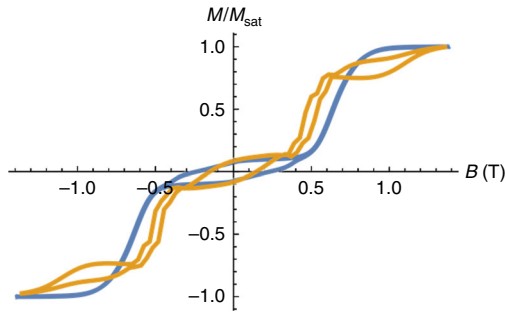

**Fig. 9** Simulation of single-crystal magnetization. Single-crystal experimental magnetization (blue curve) measured at $T = 0.03$ K and a sweep rate of 0.1 T s$^{-1}$ for a magnetic field oriented parallel to the triangle's planes and along the easy axis ($y$-axis in Fig. 6), superimposed on the simulated dynamical magnetization at the same temperature, sweep rate, and field orientation, by solving Eq. (8) on the basis of 56 out of the 256 states obtained from our model and reported in Supplementary Fig. 8, for the following numerical values of the transition rates appearing in the equations: $\Gamma_{Cr} = 10^5$ Hz/(cm$^{-1}$)$^3 \gg \Gamma_1 = 10^{-7} \times \Gamma_{Cr} > \Gamma_2 = 10^{-3} \times \Gamma_1$, and $\gamma_{Cr} = 10^{16}$ Hz$^2 \gg \gamma_1 = 10^{12}$ Hz$^2 > \gamma_2 = 10^{-3} \times \gamma_1$

fulfilling our proposed hierarchy $\gamma_{Cr} \gg \gamma_1 > \gamma_2 > \gamma_3 \sim 0$, and $\Gamma_{Cr} \gg \Gamma_1 > \Gamma_2 > \Gamma_3 \sim 0$. See methods section for a detailed discussion of the numerical choice of the parameters $\gamma_i$ and $\Gamma_i$.

We numerically solved Eq. (8) for the time-dependent populations $\sigma(t)$ of the 56 CrDy$_6$ states considered here (Supplementary Figs. 8, 10). The magnetization curve $M(t) = \mathrm{Tr}[\sigma(t)M]$ is then computed ($M$ is the magnetic moment operator along the field), parametrically plotted vs. the sweeping field, and reported in Fig. 9 with the experimental magnetization.

Quite remarkably, the opening of the hysteresis in the field region between $+B_{LC}$ and $-B_{LC}$ is captured by our model, together with the narrowing and closing of the hysteresis loop at fields $\sim B_{LC}$. Also, the simulated dynamical magnetization now displays a smooth increase between the low- and high-field "plateaus", despite the low temperature, on account of the cascade of indirect transitions between the low-field (FT) and high-field (onion) ground states mediated by intermediate excited magnetic states.

There are, of course, shortcomings in our simulation. For instance, the zero-field hysteresis loop, after closing down as in the experiment, opens up again at larger fields, a feature that is only marginally present in the experiment. Also, the theoretical magnetization step covers a smaller field range than the experimental one. While we cannot exclude that a more thorough exploration of the parameter space might improve the fitting, we expect that these shortcomings will be partially overcome if all 256 states are included in the simulation, as the additional excited states would participate to the multi-step magnetization relaxation, further widening the field range covered by the magnetization step. Further discussion of our dynamical model, including a partitioning of the contributions to $M(t)$ from individual states, is reported in the Supplementary Note 5.

In summary, a heptanuclear {Cr$^{III}$Dy$^{III}_6$} complex has been synthesized and structurally characterized. Experimental evidence in conjunction with theoretical calculations reveal and explain the presence of both SMM and single-molecule toroidic behavior. The toroidal states in the individual {Dy$^{III}_3$} triangles are found to be ferrotoroidically coupled. We note here that the Cr$^{III}$ ion does not play any fundamental role in the predicted FT coupling, and the coupling between the two toroidal wheels can in fact be fully explained solely in terms of dipolar interactions, which depend solely on the structural parameters of the complex. The fundamental structural elements influencing the strength of dipolar FT are the staggered arrangement of the two

triangles with respect to each other, and the distance $h$ between the two wheels, which we found not to be optimal for maximizing FT coupling (i.e., $h > r\sqrt{7/2}$). The design and synthesis of similar {M$^{III}$Dy$^{III}_6$} complexes, maintaining the staggered arrangement of the two triangular units, but featuring a smaller, even diamagnetic ion, is expected to lead to a smaller inter-ring distance $h$, or even to $h < r\sqrt{7/2}$, thus according to Eq. (6), to a stronger FT coupling. This route is currently being explored in our labs. Importantly, our findings indicate, for the first time, how coupling between toroidal moments can be manipulated by structural design. Finally, for the first time, the experimental single-crystal magnetization dynamics of a polynuclear Dy complex, displaying zero-field opening of a hysteresis loop, is simulated via a theoretical dynamical model, showing that the FT ground state plays a pivotal role in hindering the flipping of magnetic onion states in a sweeping field, thus slowing down the zero-field magnetic relaxation.

## Methods

**Synthesis of [Dy$^{III}_6$Cr$^{III}$(OH)$_8$(*ortho*-tol)$_{12}$(MeOH)$_5$(NO$_3$)]·3MeOH (1)**. Cr(NO$_3$)$_3$·9H$_2$O (0.4 g, 1 mmol) and Dy(NO$_3$)$_3$·6H$_2$O (0.22 g, 0.5 mmol) were dissolved in MeCN (20 ml), followed by the addition *ortho*-toluic acid (0.14 g, 1.0 mmol) and triethylamine (0.55 ml, 4.0 mmol), which resulted in a pale purple solution. This solution was stirred for 4 h after which time the solvent was removed to give a purple oil. The oil was re-dissolved in MeOH/$^i$PrOH (1:1) and the solution allowed to slowly evaporate. Within 10–15 days, pale purple crystals of 1 had appeared, in approximate yield of 25% (crystalline product). Microanalysis for CrDy$_6$C$_{104}$H$_{124}$NO$_{43}$: expected (found); C 40.25 (39.86), H 4.02 (3.86), N 0.45 (0.62).

The synthesis reaction was carried out under aerobic conditions. Chemicals and solvents were obtained from commercial sources and used without further purification.

**X-ray crystallography**. X-ray measurements for **1** were performed at 100(2) K at the Australian synchrotron MX1 beam line[45]. The data collection and integration were performed within Blu-Ice[46] and XDS[47] software programs. Compound **1** was solved by direct methods (SHELXS-97)[48], and refined (SHELXL-97)[49] by full least matrix least squares on all $F^2$ data within X-Seed[50] and OLEX-2 GUIs[51]. Crystallographic data and refinement parameters are summarized in Supplementary Table 1.

**Magnetic measurements**. The magnetic susceptibility measurements were carried out on a Quantum Design SQUID magnetometer MPMS-XL 7 operating between 1.8 and 300 K for DC-applied fields ranging from 0 to 5 T. Microcrystalline samples were dispersed in vaseline in order to avoid torquing of the crystallites. The sample mulls were contained in a calibrated gelatine capsule held at the center of a drinking straw that was fixed at the end of the sample rod. Ac susceptibilities were carried out under an oscillating ac field of 3.5 Oe and frequencies ranging from 0.1 to 1500 Hz.

**EPR instrumentation**. The X-band measurements were made on Bruker spectrometers, at IIT Bombay, with a helium gas-flow cryostat. The X-band measurements were carried out at 5, 10, and 20 K.

**Computational details**. Even though there are only three crystallographic nonequivalent Dy$^{III}$ centers, we performed the calculations on all six Dy$^{III}$ ions to determine the direction of the local anisotropy axis. Using MOLCAS 7.8[43] ab initio calculations were performed on the six Dy$^{III}$ ions using the crystal structure of **1**. The structure of the modeled Dy fragment employed for calculation is shown in Supplementary Fig. 5, where the neighboring Dy$^{III}$ ions are replaced with diamagnetic Lu$^{III}$ ions and the Cr$^{III}$ ion replaced with Sc$^{III}$ ion. We have employed this methodology to study a number of Dy$^{III}$/Er$^{III}$ SMMs[52–55]. The relativistic effects are taken into account based on the Douglas–Kroll Hamiltonian[56]. The spin-free eigenstates are achieved by the Complete Active Space Self-Consistent Field (CASSCF) method[57]. We have employed the [ANO-RCC…8s7p5d3f2g1h.] basis set[58] for the Dy atoms, the [ANO-RCC…3s2p.] basis set for the C atoms, the [ANO-RCC…2 s.] basis set for H atoms, the [ANO-RCC…3s2p1d.] basis set for the N atoms, the [ANO-RCC…4s3p1d.] basis set for the Sc atom, the [ANO-RCC…5s4p2d1f.] basis set for the La atom, and the [ANO-RCC…3s2p1d.] basis set for the O atoms. First we performed the CASSCF calculation including nine electrons across seven 4f orbitals of the Dy$^{3+}$ ion. With this active space, we have computed 21 roots in the configuration interaction procedure. After computing these excited states, we have mixed all roots using RASSI-SO[59] and spin–orbit coupling is considered within the space of calculated spin-free eigenstates. Moreover, we have considered these computed SO states into the SINGLE_ANISO[60]

program to compute the g-tensors. The $Dy^{III}$ ion has eight low-lying KDs for which the anisotropic g-tensors have been computed. The Cholesky decomposition for two electron integrals is employed throughout our calculations. We have extracted the crystal field parameters using the SINGLE_ANISO code as implemented in MOLCAS 7.8.

DFT calculations were performed using the B3LYP functional[61] with the Gaussian 09 suite of programs[62]. To estimate the exchange constant between $Cr^{III}$–$Dy^{III}$ and $Dy^{III}$–$Dy^{III}$ ions, the dysprosium ions were replaced with the spin-only $Gd^{III}$ ions in order to investigate the exchange interaction between the $Dy^{III}$ ions, which was then rescaled to the spin of dysprosium ions. We have used the LanL2DZ ECP basis set for Cr[63, 64], the double-zeta quality basis set employing Cundari–Stevens (CS) relativistic effective core potential on Gd atom[65], 6–31G* basis set for the rest of the atoms. The DFT calculations combined with the Broken Symmetry (BS) approach[66] have been employed to compute the magnetic exchange J value. The actual energy spectrum and wavefunction and magnetic texture calculations, together with simulation of the magnetic measurements, was carried out by inserting the ab initio data into a model intramolecular magnetic coupling Hamiltonian, the development of which is described in the theoretical analysis section (see above).

**Method of simulation of the magnetization dynamics.** As is well known[67], in the derivation of the Pauli equations for the diagonal reduced density matrix σ from the Liouville–Von Neumann equations for the total density matrix describing the coupled system–phonon reservoir, it is expedient to switch to the interaction picture, so that the unperturbed quantum system and reservoir Hamiltonians do not explicitly appear in the transformed equations of motion. The equations of motion in the interaction picture are thus propagated only by the spin–phonon coupling Hamiltonian, which in second order and in the Born–Markov limit generates the dissipative relaxation dynamics of the reduced density matrix. In our particular case, where time dependence arises both from the random dissipative relaxation fields, and from the periodic time dependence of the Zeeman Hamiltonian, a useful interaction picture can still be achieved by including the time-dependent Zeeman Hamiltonian in the zeroth order Hamiltonian $H_0(t) \equiv H_0$ together with Hamiltonian Eqs. (1) and (2). To avoid the complications related to the resulting time dependence of $H_0$, which would imply a zeroth order time evolution operator only defined exactly via time-ordered products, we assume valid adiabatic approximation, so that at any time the system is assumed to be described by the eigenfunctions $|i(t)\rangle$. and eigenvalues $E_i(t)$ of the time-dependent Hamiltonian $H_0$, so that $H_0(t)|i(t)\rangle = E_i(t)|i(t)\rangle$, and the time evolution operator can be simplified as an exponential operator that remains diagonal in the basis of the 256 system's eigenfunctions $|i(t)\rangle$ at any time. Using well-known approximations to avoid at each given time, the integration of eigenvalues at all previous times, as described for instance in the paper[68] and within the secular approximation[67], two decoupled sets of equations of motion are obtained for the elements of the reduced density matrix $\sigma_{ij} = \langle i|\sigma|j\rangle$ between the adiabatic eigenstates of $H_0$, one for the diagonal (populations, $i = j$) and one for the off-diagonal (coherences, $i \neq j$) elements of the CrDy$_6$ reduced density matrix σ, which can be written in the usual Schrodinger's picture as:

$$\dot{\sigma}_{ij}(t) = -i/\hbar\langle i|[H_0(t), \sigma(t)]|j\rangle + \delta_{ij}\sum_k W_{k\to i}(t)\sigma_{kk}(t) - \mu_{ij}(t)\sigma_{ij}(t),  \quad (9)$$

where $\dot{\sigma}_{ij} \equiv d\sigma_{ij}/dt$, $[H_0, \sigma]$ is the commutator between the operators $H_0$ and σ, $\delta_{ij}$ is a Kronecker delta, $W_{l\to m}(t)$ is the transition rate from eigenstate $|l(t)\rangle$ to eigenstate $|m(t)\rangle$, which become themselves time dependent because of the time-dependent Zeeman field, both via the Zeeman eigenvalues $E_i(t)$, but in principle also via the eigenstates $|l(t)\rangle$, as Cr spin states are now coupled by the magnetic field. Finally, we have $\mu_{ij}(t) = 1/2\sum_k (W_{i\to k}(t) + W_{j\to k}(t))$.

The details of the spin–phonon coupling Hamiltonian contributing to the transition rates for this polynuclear system coupled to its crystal phonons represents in principle a formidable problem, whose detailed analysis goes beyond the scope of the current paper. Hence we describe here the coupling of the CrDy$_6$ states to an idealized equilibrium acoustic phonon reservoir in terms of well-known phenomenological transition rates $W_{l\to m}(t)$ obtained within the Debye model[2] and given by:

$$W_{i\to j}^{ph}(t) = \Gamma_{ij}\frac{(E_i(t) - E_j(t))^3}{\exp[(E_i(t) - E_j(t))/k_BT] - 1}, \Gamma_{ij} = \Gamma_{Cr}, \Gamma_1, \Gamma_2,  \quad (10)$$

where $E_i(t)$ are now the time-dependent Zeeman energies of the CrDy$_6$ quantum states, while $\Gamma_{ij}$ are numerical parameters measuring the transition rate $\Gamma_{Cr}$, or $\Gamma_1$, or $\Gamma_2$, according to how many spin flips connect states $|i(t)\rangle$ and $|j(t)\rangle$ (we set $\Gamma_{ij} = 0$ if the two states are connected by three or more spin flips).

Finally, in order to account for the relaxation dynamics associated to quantum tunneling processes induced by, e.g., random stray magnetic fields produced by the fluctuations of nuclear dipole moments or neighboring molecular magnetic moments, we follow the approach proposed by Leuenberger and Loss[69] (see in particular the derivation of their Eqs. (35) and (36)), and further correct the zeroth order Hamiltonian $H_0$ in Eq. (9) with an additional term $V$ describing tunneling between the $Dy^{3+}$ and $Cr^{3+}$ magnetic states via, e.g., coupling to fluctuating magnetic fields arising from neighboring spins. We note in fact that in Eq. (9), the

diagonal matrix elements of the commutator between the eigenstates of $H_0$ are exactly zero in the adiabatic approximation, and thus the differential equations for the populations are there decoupled from those for the coherences. If on the other hand we now introduce the tunneling operator $V$, so that in Eq. (8) $H_0(t) \to H_0(t) + V$, since $V$ by definition has non-zero off-diagonal matrix elements between the eigenstates of $H_0$, this correction will re-introduce coupling between populations and coherences in Eq. (9). Assuming a steady-state approximation for the coherences, so that $\dot{\sigma}_{ij}(t) \approx 0$ on the timescale over which the populations $\sigma_{kk}(t)$ display appreciable changes, Eq. (9) can be transformed in a system of differential equations for the populations only, a set of generalized Pauli equations that reads (this is Eq. (8), reported here for convenience):

$$\dot{\sigma}_{ii}(t) = \sum_k \left\{ W_{k\to i}^{ph}(t) + \Omega_{k\to i}^{tun}(t) \right\}\sigma_{kk}(t) - \sigma_{ii}(t)\sum_k \left\{ W_{i\to k}^{ph}(t) + \Omega_{k\to i}^{tun}(t) \right\},$$

$$\quad (11)$$

where the time-dependent incoherent tunneling transition rates $\Omega_{k\to i}^{tun}(t)$ are given by:

$$\Omega_{k\to i}^{tun}(t) = \gamma_{ki}\frac{\mu_{ij}(t)}{\omega_{ij}^2(t) + |\mu_{ij}(t)|^2} \approx \gamma_{ki}\frac{\lambda}{\omega_{ij}^2(t) + \lambda^2}, \text{with } \gamma_{ki} = \gamma_{Cr}, \gamma_1, \gamma_2,  \quad (12)$$

where $\omega_{ij}(t) = (E_i(t) - E_j(t))/\hbar$. We note that Eq. (12) is in fact the equation of a Lorentzian lineshape with a maximum at the resonance $\omega_{ij}(t) = 0$, thus describing incoherent tunneling processes between Zeeman states in proximity of level crossing. While in principle this derivation predicts via Eq. (12) that the broadening of the Lorentzian lineshape should also be treated as time-dependent, we choose here to fix the broadening as a constant parameter $\lambda$ to simplify our dynamical model.

The task of solving Eq. (11) is computationally not trivial, and to make the calculations faster and more stable, instead of including all the 256 states of the CrDy$_6$ system arising from our low-energy model, we decided to include only the lowest energy states over the magnetic field range explored, which beside the 16 FT and AFT states, they correspond to Dy-based magnetic states with anisotropy axes aligned along the sweeping field. These states correspond in fact to the configurations shown in Supplementary Fig. 8. We report in Supplementary Fig. 10 the 56 Zeeman levels entering our dynamical model (Eq. (11)), as function of the magnetic field, which can be compared with the full plot of the 256 Zeeman levels in Fig. 8b–d in the main text.

Note that in principle we have at least seven free parameters entering Eq. (11): the three spin–phonon relaxation rates $\Gamma_{Cr}$, $\Gamma_1$, and $\Gamma_2$, with $\Gamma_{Cr} \gg \Gamma_1 > \Gamma_2$, the three squares of tunneling relaxation rates $\gamma_{Cr}$, $\gamma_1$, and $\gamma_2$ with $\gamma_{Cr} \gg \gamma_1 > \gamma_2$, and the Lorentzian broadening $\lambda$. The actual maximal relaxation rate at the Lorentzian maximum (i.e., at exact level crossing) is in fact given by $\gamma_k/\lambda$, with $\gamma_k$ corresponding to the square of the tunneling splitting at level crossing expressed in Hz. Some of these parameters can in fact be fixed within reasonable ranges. For instance, from ref. [70]., we learn that the typical range of values for $\Gamma_{Cr}$, i.e., the spin–phonon relaxation rate for simple paramagnetic ions, is in the range $3 \times 10^3$–$3 \times 10^5$ Hz/(cm$^{-1}$)$^3$. We further assume that fluctuating stray fields are of the order of 1 mT, given that the magnitude of the non-fluctuating dipolar field induced at any $Cr^{3+}$ or $Dy^{3+}$ site within one molecule varies between ~8 mT (inter-wheel interactions) and ~200 mT (intra-wheel interactions), and that due to the strongly anisotropic and non-collinear character of the local magnetic moments, orientational effects will greatly reduce these fields, which can only induce tunneling when oriented perpendicular to the local anisotropy axes. Considering that the transition magnetic moment between $Cr^{3+}$ spin states whose $M_S$ quantum number differ by one unit is ~$2\mu_B$, we get the following rough estimation for $\gamma_{Cr} \sim (2\mu_B \times 1mT)^2/\hbar^2 \sim 10^{16}$ Hz$^2$ (corresponding to a maximal tunneling frequency in the absence of broadening of ~0.1 GHz). Also, from our single-ion ab initio calculations for the $Dy^{3+}$ ions in CrDy$_6$, we find that the average value of the transition matrix element between the two components $M_J \sim \pm 15/2$ of the ground KD on each ion is of the order of $\sim 10^{-2}\mu_B$. Given that the 1-flip tunneling transitions only involve one $Dy^{3+}$ ion at the time, we obtain $\gamma_1 \sim (10^{-2}\mu_B \times 1mT)^2/\hbar^2 \sim 10^{11} - 10^{12}$Hz$^2$ (i.e., a maximal tunneling frequency of 1 MHz). While these rough considerations reduce the possible values for three of the seven free parameters, we still need to arbitrarily fix four of them: $\Gamma_1$, $\Gamma_2$, $\gamma_2$, and $\lambda$.

Given the approximate nature of the model, it is not our aim to attempt a fully satisfying fitting of the remaining four parameters. Thus guided by the relations $\Gamma_{Cr} = 10^5$ Hz/(cm$^{-1}$)$^3 \gg \Gamma_1 > \Gamma_2$, and $\gamma_{Cr} = 10^{16}$ Hz$^2 \gg \gamma_1 = 10^{12}$ Hz$^2 > \gamma_2$, we found that a reasonably good agreement between the calculated magnetization $M(t) = \text{Tr}[\sigma(t)M]$ and experimental magnetization can be obtained for $\Gamma_1 = 10^{-7} \times \Gamma_{Cr}$, $\Gamma_2 = 10^{-3} \times \Gamma_1$, $\gamma_2 = 10^{-3} \times \gamma_1$, and $\lambda = 10^{10}$ Hz, which are the parameters used to obtain both Fig. 9 and Supplementary Fig. 11.

**Data availability.** The X-ray crystallographic coordinates for structure reported in this study have been deposited at the Cambridge Crystallographic Data Centre (CCDC), under deposition number 1435033. These data can be obtained free of charge from the Cambridge Crystallographic Data Centre via www.ccdc.cam.ac.uk/data_request/cif.

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

## Acknowledgements

G.R. would like to acknowledge the financial support from DST-India, and IIT Bombay for the high performance computing facility. K.S.M. and G.R. thank the Australia-India AISRF program for support. K.R.V. is thankful to the IITB-Monash Research Academy for a PhD studentship. G.R. acknowledges funding from SERB-DST (EMR/2014/000247) for financial support. A.S. acknowledges support from the Australian Research Council, Discovery Grant ID: DP15010325. W.W. acknowledges the Alexander von Humboldt foundation

## Author contributions

A.S., K.S.M., S.K.L., and G.R. visualized and designed the project. K.R.V. and S.K.L. carried out the syntheses and characterized the materials. S.K.L. and K.R.V. performed the synchrotron X-ray scattering measurements, analyzed the data, and solved the crystal structure. W.W. carried out micro-SQUID measurements. K.R.V. and A.S. carried out the ab initio calculations. A.S. developed the theoretical models for magnetic coupling, and for the hysteretic dynamics of the magnetization, and used them for the simulation and interpretation of the magnetic data. A.S., K.S.M., and G.R. wrote the manuscript. All the authors discussed the results and contributed to the manuscript.

## Additional information

**Competing interests:** The authors declare no competing financial interests.

