## [Peer Review File · Nature Communications]

Reviewers' Comments:

Reviewer #1:

Remarks to the Author:

This manuscript reports the synthesis and the magnetic and structural characterization of a CrDy₆ molecular nanomagnet constituted by two Dy₃ triangular units connected through a Cr ion. By using ab-initio calculations, the Authors predict toroidal states for the two triangular units and a ferrotoroid coupling between them. This model is supported by the favorable comparison between the corresponding simulations and magnetometry and EPR measurements. Additionally, ac susceptibility and very low temperature magnetization measurements are exploited to investigate the slow relaxation dynamics.

As stated in my previous report, the study of complexes with a toroidal arrangement of local magnetic moments is not new (see e.g., Refs [8-14,21, 42-45]), but CrDy₆ is the first one in which a ferrotoroid ground state is suggested. This makes CrDy₆ a very interesting molecule from the point of view of fundamental research and possibly for future applications in quantum information or in the design of molecule-based multiferroics. Hence, these results are interesting for people working in molecular magnetism. Moreover, the Authors have now considerably strengthened the manuscript by adding micro-squid and EPR measurements. However, there are still a few points that need to be addressed. Hence, I recommend this work for publication in Nature Communications if the Authors address the following remarks.

-The Authors should state more clearly to which extent the experimental data demonstrate the validity of the model deduced from ab-initio calculations. It is remarkable that a model in which all the parameters are calculated ab-initio well explains powder magnetization, susceptibility and EPR data. However, in present version of the paper it is not clear to which degree a different model (not characterized by toroidal states) can be excluded from experimental data.

-The discussion on single-crystal magnetization measurements should be significantly expanded. Are the positions of the observed steps in quantitative agreement with the model? Do these low-T data enable one to rule out a non-toroidal nature of the lowest-energy states? It would be useful to add in the Supplementary Information the calculated field dependence of the low-lying energy levels of Dy₆Cr corresponding to the two measured orientations.

-Reporting the simulated EPR spectrum in the main paper would help the reader in judging the validity of the model. In addition, it would be helpful to add a comment on the fact that the high-field part of the experimental spectrum is much broader than the simulation. Is the measured T-dependence of the spectrum in agreement with the model?

-The theoretical analysis of magnetic relaxation is not clear (for instance, the Authors need to justify the way used to calculate the probability of thermally assisted tunneling) and does not enable to draw sound conclusions. Indeed, the Authors conclude "This suggests that other factors are involved and the magnetic blocking does not arise simply from the single ion DyIII anisotropy". Moreover, it does not add much to the main story. I think that the Authors should remove this part or make it much clearer.

-The sentence in the abstract "the split ferrotoroidic and antiferrotoroidic quantum states resulting from the con-rotating and counter-rotating coupling of local toroidal states on single triangular units can be useful to implement quantum gates between toroidal spin qubits" needs to be discussed and justified in the main text.

Reviewer #2:

Remarks to the Author:

Combining various experimental techniques and a profound theoretical analysis, the manuscript

presents an interesting example of a novel magnetic order in 3d-4f complexes. The connection of two Dy₃ triangles via a Cr(III) ion is shown to give rise to a ferrotoroidic magnetic state. The synthesis and structural characterization are described in a concise manner and the subsequent description of the magnetic properties establishes the presence of toroidal magnetic moments at low temperatures ($\leq 3\text{K}$) and gives evidence for SMM behaviour albeit with a low blocking temperature. The theoretical analysis is based on the construction of N-electron states for the Dy₆Cr system using single-ion anisotropy calculations combined with isotropic estimates of the exchange interactions between the ions. The findings of these mononuclear or binuclear calculations are used to construct the spin states of the heptanuclear cluster and investigate the relative order of the different spin orderings. The results of the model Hamiltonian study are in good agreement with the experimental findings and provide additional evidence for a ferrotoroidic ground state. The system turns out to be a nice example of how one can combine a single DyIII triangle into a larger system with larger net moments, and hence, can be expected to inspire other researchers to design similar systems with stronger coupling and possibly improved SMM properties. The paper can be published although a few points could be changed to improve the presentation of the result as specified below in arbitrary order.

Some sentences are not as crystal-clear as they could be. For example, the first sentence of the second paragraph of the introduction is rather cryptic: "A key property...intramolecular magnetic coupling". I find it difficult to understand what is meant by this sentence. Is the Zeeman effect smaller than the intramolecular magnetic coupling? A reformulation of the phrase may help the (non-specialist) reader to get the picture. A second example is the one-but-last sentence of the left column on page 4. The procedure followed involves the calculation of the relative energies of the exchange coupled states by diagonalization of the Heisenberg Hamiltonian in the basis of the lowest KDs using isotropic J's instead of a more rigorous (but obviously too complicated) treatment with real anisotropic J_{ij} coupling parameters. This is not obvious from the formulation in the manuscript.

The discussion of the magnetic relaxation mechanism does not seem all that relevant. The suggested relaxation involving the first excited state gives far too large a barrier. As the authors mention other factors are involved. But the computational results suggest that neither of the proposed mechanisms (ground state tunneling and relaxation via excited state) are relevant. This paragraph could be suppressed.

In the discussion of the calculated g-values, it is stated that the different coordination of the Dy1 and Dy1' ions (MeOH versus nitrate) is reflected in the calculated g-values. Looking at the data in the supplementary information, this is only a rather subtle difference and not really significant to be explicitly mentioned in the main text.

Quite some discussion is made of the possibly different crystal structure for the ferro and antiferrotoroidic coupled states triggered by the incompatibility of an inversion center and a ferrotoroidic magnetic ordering. Do the authors really think that the distortion can ever be large enough to be detected?

The Hamiltonians given in eq 1 and 2 are diagonal in the chosen basis by the definition of the spin operators and not by the large projection of the ground state angular moments in the $|m, M_{\text{Cr}}\rangle$ functions. Eq 1 is always diagonal and Eq. 2 becomes diagonal in any spin basis when the total spin operator is replaced by the tangential component of the spin moment as is done by the authors. The fact that the chosen basis has a large projection on the ground state angular moments indicates that the basis functions are good representations of the lowest electronic states and that the eigenvalues of the Hamiltonians accurately represents the relative energies. The authors may consider to replace the expression "it can easily be seen" by a more explicit formulation. Given the fact that the journal aims for a relatively broad audience, this will probably be more useful for people that do not work with spin Hamiltonians on a daily basis.

Reviewer #3:

Remarks to the Author:

The manuscript describes a detailed investigation of a new type of heterometallic cluster that comprises two Dy₃ triangles, which exhibits toroidal arrangement of the magnetic moments. Though toroidal ground state for Dy triangles has already been reported the authors claim here that due to the fact that dipolar interaction overwhelms exchange one and stabilizes a state where the two triangles rotation of the magnetic moment.

The subject is interesting and the manuscript potentially suitable for Nature Communications. However the authors seem to have missed a previous article where two Dy₃ triangles are ferromagnetically coupled and the same "ferrotoroidal" order is obtained, though through AF interaction (see Hewitt, I. J. et al. *Angew. Chem. Int. Ed.*, 2010, 49, 6352). Moreover in that case it seems that the ferrotoroidal ground state is more stabilized and actually shows an improved SMM behavior, which is indeed the principal goal of the strategy developed by the authors of the present manuscript.

Without a proper reference and comparison with existing literature an assessment of this manuscript cannot be accomplished and a revised version need to be evaluated.

Other critical points are:

- Why the authors do not simulate also the microSQUID magnetization curves. Though the absolute value of these measurements the ratio between the first small saturation at ca. 0.2 Msat and the Msat observed at 1.0 T will provide information about the nature of the small magnetic moment at low magnetic field (only Cr³⁺ spins or also contribution from the Dy³⁺ magnetic moment). In principle the ferrotoroidal state, if the system is centrosymmetric, should provide zero contribution. In the AFT state also the deviation from the in-plane anisotropy of the Dy should also contribute.
- The simulation of the EPR spectrum should be reported in figure 4a
- The tunnel probability inside the ground doublet should not be indicated as small as it is indeed quite large, or not significant smaller than that involving excited states. By the way authors should specify what quantity are calculating and with what Hamiltonian as in principle in a Kramers the doublets are degenerate and not admixed.
- Page 4, end of 1st column. The sentence about the FT state not being compatible with the inversion symmetry is not clear to me.

Reviewer #4:

Remarks to the Author:

I have been asked to review this revised version of a manuscript originally submitted to Nature Chemistry, which is now being considered for publication in Nature Communications. Whilst the authors have addressed some of the concerns of the four referees I am not completely satisfied with some of their responses. Two referees point out that some systems already existing in the literature have not been properly acknowledged. Indeed, if the authors had read chapter 4 "Single-Molecule Toroids and Multinuclear Lanthanide Single-Molecule Magnets" in the book on "Lanthanide Single Molecule Magnets" by J. Tang and P. Zhang (Springer 2015) they would have understood this point better. In particular, the Dy₆ system of coupled triangles, reported in 2010 by Hewitt et al in *Angewandte Chemie (Int. Ed. Reference, 2010 49, 6352–6356 "Coupling Dy₃ triangles enhances their slow relaxation")* shows a ferrotoroidal arrangement even if the word "ferrotoroidal" was not specifically coined. That thorough study included single crystal measurements and ab initio calculations. Tang and Zhang also point out that another Dy₆ system of coupled triangles reported by Murugesu et al: *Chem Commun.* 2009, 1100 – 1102, "Linking high anisotropy Dy₃ triangles to create a Dy₆ single- molecule magnet" also appears to show a similar ferrotoroidicity, although in this case only the magnetic data can be used to support this identification.

Thus the claim from the authors that:

“More importantly, we find, for the first time, a ferrotoroidically coupled ground state, consisting of two con-rotating toroidal moments localised on the {DyIII } triangular rings, thus leading to an overall enhanced toroidal magnetic moment in the ground state for the {CrIIIDyIII } complex” as well as their remarks in the concluding part of the discussion

is not actually justified.

The authors also state in response to referee 1: Ferrotoroidicity offers a chance to enhance the toroidal moment compared to single molecule. As the systems gets larger and larger (such as Dy₆ wheels for example), it is often difficult to control the orientation of the anisotropy resulting in either the absence of toroidal moment or mixed toroidal moments. The use of triangular units as building blocks maintains a local toroidicity, and the implementation of “ferromagnetic coupling” between the blocks as shown here, is expected to achieve large toroidal moments more readily than in single molecules. More interestingly, quantum states resulting from the coherent coupling of two well-localised sets of toroidal states could be more useful to quantum computation than a single large toroidal moment, if e.g. a way to control coherent tunnelling dynamics between ferrotoroidic and antiferrotoroidic states could be devised. A single (albeit larger) toroidal state would not offer such opportunity. We have now written a sentence to clarify this point in the revised manuscript.

So, the context is understood, but the literature is not properly referenced. This nice explanation should be written into the text of the manuscript with the appropriate literature references.

Note that in their reply to referee 3, the authors are assuming that reference regarding ferrotoroidicity is being made to the paper by Novitchi et al (this was reference 10 and is now reference 11) whereas it is actually the completely missing reference to Hewitt

et al (given above) which is being discussed by the referees. Therefore the answers given are irrelevant to the objections of referee 3:

“As far as the novelty of the phenomenon it is rather surprising that these authors, working on a triangle of Dy(III) ions, have inadvertently missed or intentionally neglected contributions of other groups that are now several years old.... A survey of the literature evidences that coupling of triangles with the same toroidal chirality was already achieved by the same group, investigated in detail and the published work highlighted important effects on the magnetization dynamics. ... The manuscript needs to be completely rewritten to place it correctly in the frame of the existing knowledge and reported literature.”

In regard of the Novitchi et al reference, I fail to understand why the authors do not explain to the readers that in order to achieve a ferrotoroidal state it is necessary to apply a magnetic field in that case rather than just explaining this to the referees! This is a finding in support of the current results, not detracting from them. It is also an example of including a 3d spin in the framework of toroidal arrangements of Dy₃ triangles, and therefore of relevance to the current story.

Although the authors claim to have checked through the inconsistencies in the references, they still have the paper by Soncini et al refereed to twice in references 9 and 21. Note that in reference 21 it is the German edition of Angewandte Chemie which has been cited and not, as incorrectly given the International edition. The papers are exactly the same otherwise.

So, overall I still cannot recommend this contribution for publication in its current form. The authors are again encouraged to put their work in its proper context in terms of what is already in the literature.

Reply to reviewers:

Reviewer #1:

This manuscript reports the synthesis and the magnetic and structural characterization of a CrDy₆ molecular nanomagnet constituted by two Dy₃ triangular units connected through a Cr ion. By using ab-initio calculations, the Authors predict toroidal states for the two triangular units and a ferrotoroid coupling between them. This model is supported by the favorable comparison between the corresponding simulations and magnetometry and EPR measurements. Additionally, ac susceptibility and very low temperature magnetization measurements are exploited to investigate the slow relaxation dynamics.

As stated in my previous report, the study of complexes with a toroidal arrangement of local magnetic moments is not new (see e.g., Refs [8-14,21, 42-45]), but CrDy₆ is the first one in which a ferrotoroid ground state is suggested. This makes CrDy₆ a very interesting molecule from the point of view of fundamental research and possibly for future applications in quantum information or in the design of molecule-based multiferroics. Hence, these results are interesting for people working in molecular magnetism. Moreover, the Authors have now considerably strengthened the manuscript by adding micro-squid and EPR measurements. However, there are still a few points that need to be addressed. Hence, I recommend this work for publication in Nature Communications if the Authors address the following remarks.

Reply: We thank the reviewer for appreciating our work and recommending it for publication to Nat. Commun.

1) The Authors should state more clearly to which extent the experimental data demonstrate the validity of the model deduced from ab-initio calculations. It is remarkable that a model in which all the parameters are calculated ab-initio well explains powder magnetization, susceptibility and EPR data. However, in present version of the paper it is not clear to which degree a different model (not characterized by toroidal states) can be excluded from experimental data.

Reply: To address the Referee's request we varied one of the key results of our CASSCF-RASSI-SO calculations, which plays a crucial role in determining the ferrotoroidic ground state: the direction of the local anisotropy axes of the Dy ions. From our calculations these axes turn out to be almost exactly contained in the two triangles' planes (average out-of-plane deviation been about 3°), and directed along the local tangent to the wheel's circumference (average in-plane deviation being less than 1°). To set up models that depart from this ab initio result, we generalized our exchange+dipolar coupling Hamiltonian introducing two angles: an angle η measuring the departure of the anisotropy axis from an in-plane configuration, and an angle ϕ measuring the departure of the in-plane projection of the anisotropy axis from a locally tangential direction. To comply with the D_{3d} pseudo-symmetry of the metal core of the complex, we demanded that the angle ϕ be the same for all Dy ions, while the angle η should have

opposite signs for the two wheels, due to inversion symmetry. We explored two significant scenarios departing from our parameter-free ab initio model, and reported the resulting powder magnetization curves obtained at 2K in the figure below, together with the results of our parameter-free ab initio model (orange curve in the picture) and the experimental data points (blue data points in the picture):

(i) $\eta = 30^\circ$, $\phi = 0^\circ$, *i.e.* a significant departure from in-plane tangential configuration of the magnetic axes, which will determine a significant out-of-plane magnetic moment for the Antiferrotoroidic (AFT) configuration only, but a zero out-of-plane magnetic moment for the Ferrotoroidic (FT) configurations. Such out-of-plane magnetic component of the AFT state will also be coupled antiferromagnetically to the Cr magnetic moment, thus stabilizing the AFT with respect to the non-magnetic FT state. For $\eta = 30^\circ$, the appearance of a significant anisotropic out-of-plane magnetic moment in the AFT state, determines a strong Cr-Dy₆ antiferromagnetic stabilisation energy contribution which makes the AFT configuration the ground state, and the FT state the first excited state. However, the powder magnetization we calculate in this scenario is reported in the picture below (green curve), and evidently it does not match the experimental data, which instead support our finding that at low field the only source of magnetic response comes from the Cr ion. Any additional (anisotropic) magnetism from the Dy-triangles would make the low-field magnetization steeper than what observed experimentally, which supports our finding of the FT configuration (implying a zero magnetic moment on the wheels) as the ground state.

(ii) $\eta = 0^\circ$, $\phi = 90^\circ$, *i.e.* the axes are still perfectly in-plane (contained in the planes defined by the two triangles), but they are now directed radially instead of tangentially to the triangle's circumference. In such configuration it is still possible to achieve a non-magnetic non-collinear ground state on the Dy wheels, for which the magnetism solely arises from isotropic paramagnetic Cr. However, as first pointed out by some of us in M. Giansiracusa et al. Inorg. Chem. 2016, in radially configured anisotropy axes, pure dipolar coupling does not favour such non-collinear non-magnetic configuration of the Dy magnetic moments, and favours instead a large in-plane magnetic moment in the ground state of each wheel. Furthermore, antiferromagnetic coupling to Cr ion determines the ferromagnetically coupled state (*i.e.* the state where the in-plane magnetic moments of the two Dy triangles lie parallel to each other) as the ground state, so that in fact adopting a radial instead of a tangential configuration of the magnetic axes leads to a strongly magnetic and strongly in-plane anisotropic ground state, while the states in which the two triangles have zero magnetic moment are the highest in energy. This simple rotation of the Dy anisotropy axes, still compliant with the system's pseudo-symmetry, and still allowing for the existence of non-magnetic states on the Dy triangular wheels, leads to a dramatically different exchange and dipolar coupled spectrum for CrDy₆ complex from the FT ground state predicted by our parameter-free model. However, due to the large and strongly anisotropic magnetism arising from this scenario, the low-field powder magnetization is in fact dramatically different from that experimentally observed, as can be seen from its plot in the figure below (red curve). This model also suggests that the ab initio calculations accurately reproduce the direction of the local anisotropy axes as in-plane tangential, thus stabilizing a ferrotoroidic ground state in which the magnetism solely arises from the Cr spin.

We believe that this extended model, together with the fact that our proposed model is parameter-free, only relying on experimental information (i.e. geometry of complex) and ab initio calculations, provide strong evidence that the ground state of the title compound CrDy6 is indeed ferrotoroidically coupled.

Furthermore in the new revised version we introduce a substantial extension of our discussion of the dynamics of the magnetisation in this system as observed from the single crystal magnetization experiments also introducing a theoretical model of the spin dynamics based on our model Hamiltonian which allows us to simulate and reproduced the zero-field hysteretic magnetic response observed in the experiments. This point is further discussed below but we believe it provides further evidence for the validity of our conclusions

2) The discussion on single-crystal magnetization measurements should be significantly expanded. Are the positions of the observed steps in quantitative agreement with the model? Do these low-T data enable one to rule out a non-toroidal nature of the lowest-energy states? It would be useful to add in the Supplementary Information the calculated field dependence of the low-lying energy levels of Dy6Cr corresponding to the two measured orientations.

Reply: To expand the discussion of the single-crystal magnetization measurements we made a significant expansion of the manuscript so as to include and discuss:

(i) A plot of the exchange/dipolar coupled energy levels as function of magnetic field made using our model

(ii) A plot of the equilibrium magnetization as function of field at the lowest temperature for which the measured single-single crystal magnetization was taken

(iii) A dynamical model, based on a quantum master equation reduced to the incoherent tunneling regime based on our model Hamiltonian, so to simulate the time-dependent non-equilibrium magnetization and discuss the observed hysteresis loop. This is the first time such simulations of such hysteresis loops have been reported.

All these new results are briefly discussed in the revised manuscript and elaborated further in the ESI.

3) Reporting the simulated EPR spectrum in the main paper would help the reader in judging the validity of the model. In addition, it would be helpful to add a comment on the fact that the high-field part of the experimental spectrum is much broader than the simulation. Is the measured T-dependence of the spectrum in agreement with the model?

Reply: The simulated EPR spectrum is now moved to the main paper in Figure 4a. The broadness of the observed spectrum is discussed now briefly in the revised manuscript.

4) The theoretical analysis of magnetic relaxation is not clear (for instance, the Authors need to justify the way used to calculate the probability of thermally assisted tunneling) and does not enable to draw sound conclusions. Indeed, the Authors conclude “This suggests that other factors are involved and the magnetic blocking does not arise simply from the single ion DyIII anisotropy”. Moreover, it does not add much to the main story. I think that the Authors should remove this part or make it much clearer.

Reply: As we have now expanded the theoretical analysis further, we have now removed the analysis pertaining to single ion relaxation and the pictures are moved to the ESI as these are the basis of the polynuclear calculations performed.

5) The sentence in the abstract “the split ferrotoroidic and antiferrotoroidic quantum states resulting from the con-rotating and counter-rotating coupling of local toroidal states on single triangular units can be useful to implement quantum gates between toroidal spin qubits” needs to be discussed and justified in the main text.

Reply: We have now rewritten the abstract and significantly shortened it to meet the editorial request (150 words).

Reviewer #2:

Combining various experimental techniques and a profound theoretical analysis, the manuscript presents an interesting example of a novel magnetic order in 3d-4f complexes. The connection of two Dy₃ triangles via a Cr(III) ion is shown to give rise to a ferrotoroidic magnetic state. The synthesis and structural characterization are described in a concise manner and the subsequent description of the magnetic properties establishes the presence of toroidal magnetic moments at low temperatures ($\leq 3\text{K}$) and gives evidence for SMM behaviour albeit with a low blocking temperature. The theoretical analysis is based on the construction of N-electron states for the Dy₆Cr system using single-ion anisotropy calculations combined with isotropic estimates of the exchange interactions between the ions. The findings of these mononuclear or binuclear calculations are used to construct the spin states of the heptanuclear cluster and investigate the relative order of the different spin orderings. The results of the model Hamiltonian study are in good agreement with the experimental findings and provide additional evidence for a ferrotoroidic ground state. The system turns out to be a nice example of how one can combine a single Dy(III) triangle into a larger system with larger net moments, and hence, can be expected to inspire other researchers to design similar systems with stronger coupling and possibly improved SMM properties. The paper can be published although a few points could be changed to improve the presentation of the result as specified below in arbitrary order.

Reply: We thank the reviewer for accepting our manuscript for publication in Nat. Commun. and we have now submitted the revised version addressing all the concerns of this reviewer.

1) Some sentences are not as crystal-clear as they could be. For example, the first sentence of the second paragraph of the introduction is rather cryptic: "A key property...intramolecular magnetic coupling". I find it difficult to understand what is meant by this sentence. Is the Zeeman effect smaller than the intramolecular magnetic coupling? A reformulation of the phrase may help the (non-specialist) reader to get the picture. A second example is the one-but-last sentence of the left column on page 4. The procedure followed involves the calculation of the relative energies of the exchange coupled states by diagonalization of the Heisenberg Hamiltonian in the basis of the lowest KDs using isotropic J's instead of a more rigorous (but obviously too complicated) treatment with real anisotropic J_{ij} coupling parameters. This is not obvious from the formulation in the manuscript.

Reply: We have now rewritten the introduction and clarified the Hamiltonian employed in the revised manuscript.

2) The discussion of the magnetic relaxation mechanism does not seem all that relevant. The suggested relaxation involving the first excited state gives far too large a barrier. As the authors mention other factors are involved. But the computational results suggest that neither of the proposed mechanisms (ground state tunneling and relaxation via excited state) are relevant. This paragraph could be suppressed.

Reply: As we have now expanded the theoretical analysis further, we have now removed the analysis pertaining to single ion relaxation and the pictures are moved to the ESI as these are basis of polynuclear calculations performed. See also similar response to point 4) of Reviewer 1.

3) In the discussion of the calculated g-values, it is stated that the different coordination of the Dy1 and Dy1' ions (MeOH versus nitrate) is reflected in the calculated g-values. Looking at the data in the supplementary information, this is only a rather subtle difference and not really significant to be explicitly mentioned in the main text.

Reply: This has been corrected in the revised manuscript.

4) Quite some discussion is made of the possibly different crystal structure for the ferro and antiferrotoroidic coupled states triggered by the incompatibility of an inversion center and a ferrotoroidic magnetic ordering. Do the authors really think that the distortion can ever be large enough to be detected?

Reply: We believe it is appropriate to mention this point, not only in reply to a question raised by a previous referee, but also because the incompatibility between inversion symmetry and the magnetic texture of the ground state is known to be an important ingredient e.g. in achieving magnetically ordered phases with multiferroic properties, where the magnetically ordered structure obtained below some critical temperature is not compatible with the inversion symmetry present at higher temperatures. A well-known example is that of Cr_2O_3 and Fe_2O_3 , which, despite having the same centrosymmetric crystal structure, they have different magnetically coupled ground states hence different magnetic ordering. In particular, the magnetic ordering in Cr_2O_3 is not compatible with an inversion centre, which leads to a linear magnetoelectric effect, while magnetic ordering in Fe_2O_3 is compatible with an inversion centre, and hence it does not support linear magnetoelectric response [see e.g. D. Khomskii, *Transition Metal Compounds*, CUP 2014]. It is therefore interesting to note that our CrDy_6 system possesses an inversion symmetry but a ground ferrotoroidically coupled quantum state that, in a putative ferrotoroidically ordered phase with bulk toroidal moment τ below some critical temperature, would not be compatible with inversion symmetry, and hence would allow linear magnetoelectric response, so that application of a magnetic field B will induce an electric polarization P linear in the applied field, given by $P = \tau \times B$. This has been now been briefly clarified in the manuscript.

5) The Hamiltonians given in eq 1 and 2 are diagonal in the chosen basis by the definition of the spin operators and not by the large projection of the ground state angular moments in the $|m, M_{\text{Cr}}\rangle$ functions. Eq 1 is always diagonal and Eq. 2 becomes diagonal in any spin basis when the total spin operator is replaced by the tangential component of the spin moment as is done by the authors. The fact that the chosen basis has a large projection on the ground state angular moments indicates that the basis functions are good representations of the lowest electronic states and that the eigenvalues of the Hamiltonians accurately represents the relative energies.

The authors may consider to replace the expression "it can easily be seen" by a more explicit formulation. Given the fact that the journal aims for a relatively broad audience, this will probably be more useful for people that do not work with spin Hamiltonians on a daily basis.

Reply: This has been rewritten in the revised manuscript.

Reviewer #3:

The manuscript describes a detailed investigation of a new type of heterometallic cluster that comprises two Dy₃ triangles, which exhibits toroidal arrangement of the magnetic moments. Though toroidal ground state for Dy triangles has already been reported the authors claim here that due to the fact that dipolar interaction overwhelms exchange one and stabilizes a state where the two triangles rotation of the magnetic moment. The subject is interesting and the manuscript potentially suitable for Nature Communications.

Reply: We thank the reviewer for accepting our work to Nat. Commun., and have now carefully revised the manuscript in line with the reviewer's comment.

1) However, the authors seem to have missed a previous article where two Dy₃ triangles are ferromagnetically coupled and the same "ferrotoroidal" order is obtained, though through AF interaction (see Hewitt, I. J. et al. Angew. Chem. Int. Ed., 2010, 49, 6352). Moreover, in that case it seems that the ferrotoroidal ground state is more stabilized and actually shows an improved SMM behavior, which is indeed the principal goal of the strategy developed by the authors of the present manuscript. Without a proper reference and comparison with existing literature an assessment of this manuscript cannot be accomplished and a revised version need to be evaluated.

Reply: While we have cited the work in ref 18, we have also now performed additional calculations on these system and discussed the result in the context of our work. This is elaborated in detail in the ESI where it is pointed out the strengths and weaknesses of the Hewitt et al paper.

Other critical points are:

2) Why the authors do not simulate also the microSQUID magnetization curves. Though the absolute value of these measurements the ratio between the first small saturation at ca. 0.2 Msat and the Msat observed at 1.0 T will provide information about the nature of the small magnetic moment at low magnetic field (only Cr³⁺ spins or also contribution from the Dy³⁺ magnetic moment). In principle the ferrotoroidal state, if the system is centrosymmetric, should provide zero contribution. In the AFT state also the deviation from the in-plane anisotropy of the Dy should also contribute.

Reply: We have now simulated the microSQUID data and performed in-depth analysis. This is now included in the revised manuscript with details of the calculations given in the ESI.

3) The simulation of the EPR spectrum should be reported in figure 4a.

Reply: The simulated EPR spectrum is now moved to the main paper in Figure 4a.

4) The tunnel probability inside the ground doublet should not be indicated as small as it is indeed quite large, or not significant smaller than that involving excited states. By the way authors should specify what quantity are calculating and with what Hamiltonian as in principle in a Kramers the doublets are degenerate and not admixed.

Reply: This section has been subdued and rewritten in the context of additional simulations/relaxation mechanism discussed in the revised manuscript and elaborated in the ESI.

5) Page 4, end of 1st column. The sentence about the FT state not being compatible with the inversion symmetry is not clear to me.

Reply: The inversion symmetry operation applied to a centre carrying a spin will swap that centre with its inversion related centre, but will not change the direction of the spin moment. Hence it can be easily seen that if we invert all the spins of a clockwise rotating toroidal moment on one triangle, given the staggered arrangement of the three centers in the inversion-related triangle, clearly the toroidal moment we obtain on the inversion-related triangle will be rotating anti-clockwise. Hence only the antiferrotoroidal arrangement of magnetic moments in this system is inversion-symmetric. An ordered phase with all con-rotating toroidal moments can thus only be obtained in this crystal only if inversion symmetry is broken. If inversion symmetry were preserved in an ordered toroidal phase, the toroidal moments would have to be counter-rotating. Note that centro-symmetric systems (systems with an inversion centre) which stabilize magnetically ordered phases that break the inversion symmetry are well known, and in fact they usually lead to interesting multiferroic behaviours.

Reviewer #4:

1) I have been asked to review this revised version of a manuscript originally submitted to Nature Chemistry, which is now being considered for publication in Nature Communications. Whilst the authors have addressed some of the concerns of the four referees I am not completely satisfied with some of their responses. Two referees point out that some systems already existing in the literature have not been properly acknowledged. Indeed, if the authors had read chapter 4 “Single-Molecule Toroids and Multinuclear Lanthanide Single-Molecule Magnets” in the book on “Lanthanide Single Molecule Magnets” by J. Tang and P. Zhang (Springer 2015) they would have understood this point better. In particular, the Dy₆ system of coupled triangles, reported in 2010 by Hewitt et al in *Angewandte Chemie (Int. Ed. Reference, 2010 49, 6352–6356* “Coupling Dy₃ triangles enhances their slow relaxation”) shows a ferrotoroidal arrangement even if the

word “ferrotoroidal” was not specifically coined. That thorough study included single crystal measurements and ab initio calculations. Tang and Zhang also point out that another Dy₆ system of coupled triangles reported by Murugesu et al: Chem Commun. 2009, 1100 – 1102, “Linking high anisotropy Dy₃ triangles to create a Dy₆ single- molecule magnet” also appears to show a similar ferrotoroidicity, although in this case only the magnetic data can be used to support this identification. Thus the claim from the authors that: “More importantly, we find, for the first time, a ferrotoroidically coupled ground state, consisting of two con-rotating toroidal moments localised on the {DyIII} triangular rings, thus leading to an overall enhanced toroidal magnetic moment in the ground state for the {CrIIIDyIII} complex” as well as their remarks in the concluding part of the discussion is not actually justified.

Reply: We thank the referee for the very useful suggestions for the improvement of our manuscript, especially for having pointed out the incompleteness of our discussion of the literature concerning coupled toroidal moments. While the referee is in part satisfied with some of our previous replies addressing other referees concerns on a similar problem (but asks us to include such replies in our manuscript), the referee also points out that our discussion of the paper by Hewitt et al (Angew. Chem. Int. Ed. 2010) is completely lacking in the current version. In the revised version of our manuscript not only have we incorporated our previous comments reported in our previous reply letter about existing work in this area, in particular about the work by Novitchi et al., but we have now also included a more detailed discussion of all three main works that are particularly relevant for the study of ferrotoroidic and antiferrotoroidic states in coupled Dy₃ triangles and these details are given in the ESI as there are page limitation for Nat. Com. A comment to this effect is given in the Introduction of the revised script at the top of p. 3

2) The authors also state in response to referee 1: Ferrotoroidicity offers a chance to enhance the toroidal moment compared to single molecule. As the systems gets larger and larger (such as Dy₆ wheels for example), it is often difficult to control the orientation of the anisotropy resulting in either the absence of toroidal moment or mixed toroidal moments. The use of triangular units as building blocks maintains a local toroidicity, and the implementation of “ferromagnetic coupling” between the blocks as shown here, is expected to achieve large toroidal moments more readily than in single molecules. More interestingly, quantum states resulting from the coherent coupling of two well-localised sets of toroidal states could be more useful to quantum computation than a single large toroidal moment, if e.g. a way to control coherent tunneling dynamics between ferrotoroidic and antiferrotoroidic states could be devised. A single (albeit larger) toroidal state would not offer such opportunity. We have now written a sentence to clarify this point in the revised manuscript. So, the context is understood, but the literature is not properly referenced. This nice explanation should be written into the text of the manuscript with the appropriate literature references.

Reply: We have now cited the references as suggested by the reviewer and we have also rewritten the introduction to highlight previous relevant work.

4) Note that in their reply to referee 3, the authors are assuming that reference regarding ferrotoroidicity is being made to the paper by Novitchi et al (this was reference 10 and is now reference 11) whereas it is actually the completely missing reference to Hewitt et al (given above) which is being discussed by the referees. Therefore, the answers given are irrelevant to the objections of referee 3:

Reply: We have now cited the reference and expanded the analysis to all three systems and this is given in detail in the ESI.

5) “As far as the novelty of the phenomenon it is rather surprising that these authors, working on a triangle of Dy(III) ions, have inadvertently missed or intentionally neglected contributions of other groups that are now several years old.... A survey of the literature evidences that coupling of triangles with the same toroidal chirality was already achieved by the same group, investigated in detail and the published work highlighted important effects on the magnetization dynamics. ... The manuscript needs to be completely rewritten to place it correctly in the frame of the existing knowledge and reported literature.” In regard of the Novitchi et al reference, I fail to understand why the authors do not explain to the readers that in order to achieve a ferrotoroidal state it is necessary to apply a magnetic field in that case rather than just explaining this to the referees! This is a finding in support of the current results, not detracting from them. It is also an example of including a 3d spin in the framework of toroidal arrangements of Dy₃ triangles, and therefore of relevance to the current story.

Reply: We thank the reviewer for these suggestions. We certainly did not intentionally neglect previous work, rather we probably did not highlight it sufficiently. We have now made this point clear in the revised manuscript. As indicated above, the previous papers on coupled Dy₃ toroids had strengths and weaknesses and we have gone to the extent of making calculations on some of these to better interpret their reported data using our model (see ESI).

7) Although the authors claim to have checked through the references, they still have the paper by Soncini et al refereed to twice in references 9 and 21. Note that in reference 21 it is the German edition of Angewandte Chemie which has been cited and not, as incorrectly given the International edition. The papers are exactly the same otherwise.

Reply: Apologies. We have now removed the German edition of reference 21 and cited the correct English version of the reference.

8) So, overall I still cannot recommend this contribution for publication in its current form. The authors are again encouraged to put their work in its proper context in terms of what is already in the literature.

Reply: We thank the reviewer for the criticism that has helped us to improve the quality of the manuscript and we hope that reviewer 4, and the other reviewers, appreciate the efforts we have gone to and find the revisions acceptable for Nat. Commun.

Reviewers' Comments:

Reviewer #1:

Remarks to the Author:

The Authors have satisfactorily addressed my remarks and significantly improved the manuscript. However, the graphical quality of the new Figure 9 is below standard; in particular, it is difficult to read the horizontal axes.

Hence, I recommend the present manuscript for publication in Nature Communications after a revision of Figure 9.

Reviewer #2:

Remarks to the Author:

The authors have satisfactorily taken into account the points raised by all the reviewers. I have no further comments and the paper can be accepted for publication in its present form. Among all the new things in the revised manuscript, I am specially impressed by the magnetization curves derived with the different ad-hoc ab initio models. Very nice work!

Reviewer #3:

Remarks to the Author:

The authors have significantly improved the manuscript that should be now accepted for publication after some minor corrections that do not require further check by this reviewer.

a) In the introduction the sentence "More importantly, we find, for the first time, a ferrotoroidically coupled ground state fully determined by dipolar coupling between the two con-rotating toroidal triangles (see Supplementary Information for a detailed comparison of our findings with previous studies of coupled molecular {DyIII₃} toroids).^{10,11,18}" sounds weird.

I think it would be more transparent for readers if the authors write something like ...

Though con-rotation of spins has been previously observed in coupled molecular {DyIII₃} toroids with enhancement of magnetic bistability, the more symmetric structure of Dy₆Cr results in a better realization of ferrotoroidal arrangement (see Suppl...).

b) page 11.

Authors write "The symmetry related Dy₁', Dy₂' and Dy₃' ions possess essentially the same g-tensor."

Actually the centers are related by symmetry. Why they should be only essentially the same? More striking is that a huge difference in the level splitting is observed between Dy₁ and Dy₁'. Why? Please explain otherwise calculations seems inconsistent.

c) Caption of figure 2. It calls Fig 7 while it should be figure 8. Anyhow, I suggest to merge calculated values of FIG 8 in FIG2. The manuscript is already rather long.

Reviewer #1:

The Authors have satisfactorily addressed my remarks and significantly improved the manuscript.

However, the graphical quality of the new Figure 9 is below standard; in particular, it is difficult to read the horizontal axes. Hence, I recommend the present manuscript for publication in Nature Communications after a revision of Figure 9.

We thank the reviewer for accepting our manuscript. We have now replaced it with a high quality image for Figure 9 which is Figure 8 now.

Reviewer #2:

The authors have satisfactorily taken into account the points raised by all the reviewers. I have no further comments and the paper can be accepted for publication in its present form. Among all the new things in the revised manuscript, I am specially impressed by the magnetization curves derived with the different ad-hoc ab initio models. Very nice work!

We thank the reviewer for appreciating and accepting our manuscript.

Reviewer #3:

The authors have significantly improved the manuscript that should be now accepted for publication after some minor corrections that do not require further check by this reviewer.

We thank the reviewer for appreciating and accepting our manuscript.

a) In the introduction the sentence “More importantly, we find, for the first time, a ferrotoroidically coupled ground state fully determined by dipolar coupling between the two con-rotating toroidal triangles (see Supplementary Information for a detailed comparison of our findings with previous studies of coupled molecular {DyIII3} toroids).10,11,18” sounds weird.

I think it would be more transparent for readers if the authors write something like ... Though con-rotation of spins has been previously observed in coupled molecular {DyIII3} toroids with enhancement of magnetic bistability, the more symmetric structure of Dy6Cr results in a better realization of ferrotoroidal arrangement (see Suppl...).

We have now re-written this sentence to make this point clear.

b) page 11.

Authors write “The symmetry related Dy1', Dy2' and Dy3' ions possess essentially the same g-tensor.”

Actually the centers are related by symmetry. Why they should be only essentially the same? More striking is that a huge difference in the level splitting is observed between Dy1 and Dy1'. Why? Please explain otherwise calculations seems inconsistent.

This is mainly due to different coordination of methanol in Dy1 and nitrate in Dy1'. We explained about this in the main text.

c) Caption of figure 2. It calls Fig 7 while it should be figure 8. Anyhow, I suggest to merge calculated values of FIG 8 in FIG2. The manuscript is already rather long.

We have combined Fig.8 with Fig.2.